# Complete Genome Sequence of Ovine *Mycobacterium avium* subsp. *paratuberculosis* Strain JIII-386 (MAP-S/type III) and Its Comparison to MAP-S/type I, MAP-C, and *M. avium* Complex Genomes

**DOI:** 10.3390/microorganisms9010070

**Published:** 2020-12-29

**Authors:** Daniel Wibberg, Marian Price-Carter, Christian Rückert, Jochen Blom, Petra Möbius

**Affiliations:** 1Center for Biotechnology (CeBiTec), Bielefeld University, 33501 Bielefeld, Germany; dwibberg@cebitec.uni-bielefeld.de (D.W.); cruecker@cebitec.uni-bielefald.de (C.R.); 2AgResearch, Hopkirk Research Institute, Grasslands Research Centre, Palmerston North 4442, New Zealand; Marian.Price-Carter@agresearch.co.nz; 3Bioinformatics and Systems Biology, Justus Liebig University Gießen, D-35390 Gießen, Germany; jochen.blom@computational.bio.uni-giessen.de; 4Friedrich-Loeffler-Institut/Federal Research Institute for Animal Health, Institute of Molecular Pathogenesis, 07743 Jena, Germany

**Keywords:** nanopore-technology, de novo assembly, MAP-S lineages, virulence-associated genes, *M. avium* core and pan genome analyses, phylogeny

## Abstract

*Mycobacterium avium* (*M. a.*) subsp. *paratuberculosis* (MAP) is a worldwide-distributed obligate pathogen in ruminants causing Johne’s disease. Due to a lack of complete subtype III genome sequences, there is not yet conclusive information about genetic differences between strains of cattle (MAP-C, type II) and sheep (MAP-S) type, and especially between MAP-S subtypes I, and III. Here we present the complete, circular genome of MAP-S/type III strain JIII-386 (DE) closed by Nanopore-technology and its comparison with MAP-S/type I closed genome of strain Telford (AUS), MAP-S/type III draft genome of strain S397 (U.S.), twelve closed MAP-C strains, and eight closed *M.-a.*-complex-strains. Structural comparative alignments revealed clearly the mosaic nature of MAP, emphasized differences between the subtypes and the higher diversity of MAP-S genomes. The comparison of various genomic elements including transposases and genomic islands provide new insights in MAP genomics. MAP type specific phenotypic features may be attributed to genes of known large sequence polymorphisms (LSP*^S^*s) regions I–IV and deletions #1 and #2, confirmed here, but could also result from identified frameshifts or interruptions of various virulence-associated genes (e.g., *mbtC* in MAP-S). Comprehensive core and pan genome analysis uncovered unique genes (e.g., cytochromes) and genes probably acquired by horizontal gene transfer in different MAP-types and subtypes, but also emphasized the highly conserved and close relationship, and the complex evolution of *M.-a.*-strains.

## 1. Introduction

*Mycobacterium (M.) avium* subsp. *paratuberculosis* (MAP) is a globally important obligate pathogen of domestic and wild ruminants, causing a chronic progressive granulomatous enteritis designated as Paratuberculosis or Johne’s disease [1]. First reported at the end of the 19th century in Europe, Johne’s disease spread throughout the world, particularly in bovine dairy industries, and was later also diagnosed in sheep and goats in many countries [2]. There are strain dependent differences in disease progression in cattle and small ruminants like sheep and goats [3]. Knowledge of genomic features that convey relative virulence and pathogenicity of MAP types in different hosts is likely to help with devising control strategies and methods.

MAP can persist in the environment and has been isolated from raw milk, different non-ruminant species and humans. The evidence of a zoonotic capacity of MAP concerning Crohn’s disease in humans is under ongoing discussion [4]. Together with the obligate bird pathogens *M. avium* subsp. *avium* (MAA) and subsp. *silvaticum*, the facultative-pathogenic and genetically variable pool of organisms known as *M. avium* subsp. *hominissuis* (MAH), and the feline pathogen *M. avium* subsp. *lepraemurium*, MAP belongs to the species *M. avium* and to the *M. avium* complex (MAC) [5,6,7]. Strains of MAH cause disseminated infections, pulmonary diseases and lymphadenitis in humans, and granulomatous lesions in pigs and many other animal species with wide-ranging environmental sources of infection [8,9,10,11,12,13]. The majority of human infections occur in immunocompromised people, persons with underlying pulmonary diseases and children with cystic fibrosis [14]. In recent years, MAH represents an increasing public health concern in developed countries [15].

Based on genotyping and on phenotypic features including different host associations and growth characteristics, MAP strains have been divided into two main groups: The Cattle-type (MAP-C) also designated as type II which also includes Type B, and the Sheep-type (MAP-S), which is further subdivided into sub-groups type I, III and sub-lineages of camelid isolates [16,17]. These classifications were recently confirmed by whole genome sequence analyses (WGS) based comparisons of single nucleotide polymorphisms (SNPs) [18] in draft genomes of numerous MAP strains from diverse locations worldwide.

When trying to use genomics to understand evolution and why some strains are more virulent/pathogenic than others in different hosts, ideally multiple, completely closed genomes of each important type from diverse regions should be compared. However, whole genome sequencing and assembling of MAP is hampered by some major technical challenges. Depending on differences in media, the number of passages and the genotypes of individual strains, cultivation of these bacteria for WGS requires 4–12 weeks (MAP-C) or 4 weeks to 7 months (MAP-S) [19,20]. MAP genomes have an extremely high GC content (close to 70%) that causes biases in PCR amplification steps [21,22,23] during the preparation of Illumina libraries creating uneven read representation within the genome sequences [24,25,26]. The assembly of such regions is often inefficient resulting in gaps, if libraries are prepared following standard protocols [27]. This is further complicated by the presence of numerous repeated sequences, which make assembly even more challenging.

Currently, twelve complete MAP-C genome sequences, one human and eleven bovine-derived, from four different regions of the world, have been published. These include MAP K-10 [28], MAP4 [29], and FDAARGOS_305, which originated from U.S., MAP E1 and E93, which originated from Egypt [30], MAP/TANUVAS/TN/India08 (GenBank accession no. CP015495.1), five MAP genome sequences from South Korea (GenBank accession no. CP033909-10, CP033427-28), and MAP JII-1961 from Germany [31]. Recently, a MAP-S/type I strain (Telford) from Australia was completely closed by a combination of Illumina and PacBio sequence technology [32]. Although, there are MAP-S/type III draft genomes available—for example strain S397 with 176 contigs from U.S. [33], and strain JIII-386 with six scaffolds from Germany [34] as well as additional incomplete MAP genome sequences published—to date there are no completely closed MAP-S/type III genomes.

Results of a very recently published comprehensive pan genome and phylogenetic analysis of 28 MAP and non-MAP genomes (including MAH and MAA strains) support the prediction that the core genome evolves through SNPs and recombination events while the accessory genome is acquired by horizontal gene transfer [35]. In contrast to the non-MAP strains, the included MAP-C genomes were very stable with a low number of SNPs and accessory genes, and a lack of rearrangements [35].

Due to limitations with early versions of genomic analysis software, deciphering MAC evolution based on the presence, absence and order of genes has proven to be problematic. Major genomic differences identified as insertions or deletions among strains of MAC have been designated as large sequence polymorphisms (LSPs) [36,37]). Based on the presence of specific insertions, deletions and inversion events of LSPs, the evolutionary relationship among MAC strains was determined and a biphasic or triphasic evolution of modern MAP type strains was proposed [33,38,39]. However, six genomic insertions were found only in MAP and not in the other MAC members (LSP^P^ 2, 4, 11, 14, 15, and 16). In addition, genes of a 10 kb LSP (LSP^A^8) were absent from all MAP lineages [38]. A loss of three regions (LSP^A^4-II, MAV14, and LSP^A^18) was found to be characteristic of the MAP cattle lineage, and two deletions (LSP^A^20 (deletion #1), and deletion #2) were specific for the MAP sheep lineage [34,38]. Based on all these analyses and specific in silico observed inversions, sheep type isolates seem to be an intermediate in the evolution of MAP-C type strains by the emergence from a pathogenic clone (proto-MAP) which was closely related to MAH or *Mycobacterium intracellulare* strains [33]. Another study [34] challenged the suggestion that *M. intracellulare* was an ancestor of proto-MAP and explored how well new genomic evidence fit with the previous alternative hypotheses suggesting either evolutionary division of proto-MAP into MAP-S and MAP-C or a successive evolution from such a clone via MAP-S to MAP-C. Currently there is no clear evidence that an ancient MAH subspecies was a progenitor of MAP [35].

Genomic differences between MAP-C and MAP-S-type isolates are also evident but not fully defined due to the lack of finished MAP-S genome sequences of isolates from different origins. Based on three sheep strain draft genomes (MAP-S/type III) from the U.S., Bannantine et al. [33] proposed ten probable MAP-S specific LSP*^S^*s, which could not be detected in strain K-10 (MAP-C). Using alternative analytical techniques and including an additional unfinished sheep strain from Germany, Möbius and colleagues [34] disputed half of these suggested LSP*^S^*s because they found four putative MAP-S specific regions were fully and one partly detected in MAP K-10, MAP4 and JII-1961 genomes (all MAP-C). This study extended, merged and re-defined the previous MAP-S specific described regions [38] and [33] into: LSP*^S^* I to LSP*^S^* IV. In addition, genes that were apparently absent in sheep type strains from the U.S. and defined as a MAP-S specific deletion [deletion s∆−1] [40], were later identified in MAP-S strains from Australia and Germany [34]. Former studies discussed the existence of a third major lineage within MAP besides the cattle and the sheep type group (MAP-C and MAP-S): the type III or intermediate group, exhibiting differences regarding host association and growth characteristics [16]. Phylogenetic analysis based on sequence comparisons of *gyrA*, *gyrB*, and *recF* [41], or of genome wide homologous CDS (conserved at both, the nucleotide and amino acid level) and ncRNAs [34] revealed the subdivision of MAP-S strains into sub-groups. In addition to the previously repeatedly characterized type I, and III lineages [39], there exists a third sub-lineage, comprising MAP genomes of isolates from Arabian camelids (JQ5 and JQ6; [17]). As mentioned above, this MAP-S subdivision was clearly confirmed based on SNP analysis from whole genome sequences of 20 MAP-S and numerous MAP-C strains of different origins worldwide, presented in a large phylogenetic tree [18]. This tree emphasized the higher diversity of MAP-S relative to MAP-C strains. Currently there is a knowledge gap about type I and type III specific insertions/deletions of genes and gene clusters.

For the present work, the novel Nanopore-long sequence read technology was employed to completely close the genome of German MAP-S/type III strain JIII-386 using previously obtained Illumina whole-genome shotgun sequence data to polish the Nanopore derived sequence. Nanopore is relatively inexpensive technology and is especially promising for high GC-content WGS, since no PCR amplification is required for the preparation of Nanopore sequencing libraries. Nanopore sequencing technology, developed by Oxford Nanopore, is one of the latest DNA sequencing methods and represents a third-generation approach. Using Nanopore sequencing, a single molecule of DNA is transported through a nanometer large pore [42]. These nanopores consist of recombinant proteins embedded in a polymer membrane. A bias voltage is applied across the membrane. Nucleotides that passes through the nanopore create voltage changes that are specific for each of the four nucleotides, enabling the DNA sequence to be read out. Single DNA molecules longer than a megabase can be sequenced using Nanopore, but the resulting sequence has a rather high error rate (usually in the 5–20% range). By means of sequence assembly and polishing of the consensus sequence with high depth Illumina reads, a similar error rate in comparison to short read data can be reached [43] but provides sequence over repeat regions and other regions that are not well defined with short read methods.

The completely closed and annotated genome sequence of JIII-386 is described here. Many genomic features of JIII-386 are compared to that of the recently published closed MAP-S/type I strain Telford, the MAP-C reference genome K-10, and the draft genome of MAP-S/type III strain S397. These searches include comparisons of the gross genomic structures, numbers of SNPs and INDELs, presence of specific genomic elements, selected putative virulence associated genes, secondary metabolite and antibiotic resistance gene clusters, and the content and localization of genomic islands. Furthermore, the phylogenetic relationship of a selection of MAC genomes including available and recently published complete MAP-C, MAA, and five closed MAH genomes of different geographic origin has been calculated and presented. MAP-S specific LSP*^S^*s and deletions are examined based on this enlarged strain panel. Comprehensive core and pan genome analyses assisted in clarifying and confirming differences identified previously to define typical and possibly new characteristics of MAP-S and MAP-C strains and especially of MAP-S/type I and type III genomes within the MAC based on new finished genome sequences.

## 2. Materials and Methods

### 2.1. Strain Cultivation and DNA Isolation

The sheep strain JIII-386 was isolated in 2003 from ileal mucosa of a sheep originating from a migrating herd in North-Rhine-Westphalia in Germany. The animal did not show clinical symptoms. Paratuberculosis was suspected because of positive serological results and patho-morphological as well as histological indications after necropsy. JIII-386 was originally isolated on modified Middlebrook 7H11 solid medium (Difco) containing 10% OADC, Amphotericin B, and Mycobactin J (Allied Monitor, Fayette, AL, USA) and belongs to the strain collection of the Friedrich-Loeffler-Institut in Jena (Germany). The isolate was characterized by RFLP, MIRU-VNTR, and SSR molecular genotyping as described in detail by Möbius et al. [44]. Sub-cultivation for resequencing from cryo-preservation was done in two steps: Six weeks in Middlebrook 7H9 bouillon containing PANTA and Mycobactin J, and then four months on solid Herrold’s Egg Yolk Medium (HEYM) supplemented with Mycobactin J. Identity of the strain was confirmed by different PCR assays and MIRU-VNTR analysis according to Möbius et al. [44]. RNA free bacterial genomic DNA was extracted using a modification of the cetyltrimethyl-ammonium bromide method [45] which included digestion with RNAase A and additional wash steps.

### 2.2. Nanopore Library Preparation and MinION^®^ Sequencing

The MinION sequencing library from strain JIII-386 genomic DNA was prepared using the Nanopore Rapid DNA Sequencing kit (SQK-RAD04) according to the manufacturer’s instructions with the following changes: The starting amount of DNA was increased from 50 ng to 800 ng and an AMPure XP bead clean-up was carried out after transposon mediated fragmentation. The average fragment size was 2519 bp (n50 length: 16,846 bp). Sequencing was performed on an Oxford Nanopore MinION Mk1b sequencer at the CeBiTec (Center for Biotechnology), Bielefeld University (Bielefeld, Germany), using an R9.5 flow cell, which was prepared according to the manufacturer’s instructions.

### 2.3. Base Calling, Reads Processing, Assembly, Quality Control, and Deposition of New Genome Sequence

MinKNOW (v1.13.1) was used to control the run using the 48 h sequencing run protocol; base calling was performed offline using albacore (v2.3.1). The assembly was performed using canu v1.6 [46], resulting in a single, circular contig. This contig was then polished with Illumina short read data that originated from the initial sequencing of this strain (NCBI BioProject PRJNA390765; [34]) using Pilon [47], which was run for eight iterative cycles. Bwa-mem 0.7.12 [48] was used for Illumina read mapping for the first four iterations and bowtie2 v2.3.2 [49] for the second set of four iterations.

Completeness and assembly quality were estimated with BUSCO (v3.0.2, [50]), using the bacteria-specific single-copy marker genes database (odb9). The BUSCO analysis identified 143 of 148 core bacterial genes. This result emphasizes the completeness and quality of the genome assembly.

Genbank accession number: The new version of whole-genome sequence of *Mycobacterium avium* subsp. paratuberculosis strain JIII-386 has been deposited at DDBJ/EMBL/Genbank under the Bioproject PRJNA389670 (identical to the project number for the older version of the sequence) with accession number CP042454. The strain is available from the Leibniz-Institut, German Collection of Microorganisms and Cell Cultures (DSMZ, Braunschweig, Germany) under the accession form no.4457.

### 2.4. Genome Annotation and Comparative Analysis

The final chromosome of sheep strain JIII-386 was annotated by means of the NCBI prokaryotic genome annotation pipeline (PGAP) [51]. In addition, WebMGA was applied for COG annotation with default settings (E-value threshold of 1 × 10^−20^). tRNAs and rRNA are detected by means of RNAmmer 1.2, aragorn 1.2.38 and tRNAscan-SE 2.0 and their included RNA models. Furthermore, ncRNAs were identified and annotated by Infernal 1.1.3 and Rfam (version 14.1) implemented as profile stochastic context-free grammar called “covariance model”. The final circular plot was exported from GenDB 2.0 [52].

For structural comparative analysis, the multiple genome alignment system Mauve 2.4 [53] and the reference contig arrangement tool r2cat 1.0 [54] were applied with default settings. Island Viewer 4 helped with the detection of genomic islands [55]. This method is based on an integrative approach that employs IslandPick, SIGI-HMM, IslandPath-DIMOB, and Islander software. In addition, the genomes were compared by Geneious software to identify homologous genes of GIs that were present in some strains but not recognized by the Island viewer software. MAP-S specific regions and selected MAP virulence genes were identified by means of EDGAR 2.0 [56] and additional manual curation. Transposable elements were detected with ISfinder (https://isfinder.biotoul.fr/) with a sequence identity cut-off of >70%. In addition, annotation results were validated by comparison to other NCBI annotated genomic sequences in Geneious (Version 11.1.5). PHASTER [57] was used to identify and annotate prophage sequences in the different *M. avium* genomes. The bacterial version of antiSMASH 5.1.2 was used for the detection of secondary metabolite clusters [58]. CRISPRCasFinder (https://crisprcas.i2bc.paris-sacley.fr) was applied to detect clustered regularly Interspaced Short Palindromic Repeat (CRISPR)-Cas regions. Selected MAP genomes were also checked for antibiotic resistance genes by means of BLASTp against the ARG-ANNOT database [59]. Results of many analyses were manually inspected using Geneious Prime. Moreover, the identity of all genes that are discussed in this study was manually validated by checking and adjusting with the help of BLAST, BLASTx, PFAM, COG, and PRIAM, as described recently [60,61]. By means of genome-to-genome distance calculator (GGDC) 2.1 [62,63], the relation between two strains was calculated concerning the assignment to one species or subspecies. For phylogenetic analysis, FastTree 2.1 was used for building maximum-likelihood trees for large alignments [64]. Comparative analyses, including core and pan genome analyses, and phylogenetic classification of strains were performed with EDGAR 2.0 [56].

Comparative genome analyses have some slight limitations with respect to paralogous genes. The EDGAR platform uses bidirectional best hits (BBHs) with a BLAST score ratio values (SRVs) based threshold [65]. In short, the SRV approach normalizes all BLAST scores in relation to the best possible hit, the self-hit of a gene. One drawback of the approach used in EDGAR is that only one-to-one orthologous pairs are found. For duplicated genes and also paralogs, a single hit will be found, and additional copies are missed. However, the BBH calculation is straightforward and therefore fast enough to handle the huge amounts of sequence information in comparative genomics. As the bias due to paralogous genes appearing in different order during the calculation is usually quite small (way below 1%) [66], this is an acceptable trade-off between speed and accuracy. All genes that were annotated as pseudogenes were also neglected within these analyses.

Appendix A lists the bioinformatic tools that were used for these analyses.

### 2.5. Reference Genome Sequences

Altogether, 24 *M. avium* genome sequences, which were all annotated with the PGAP pipeline and deposited in the RefSeq database, were involved with different aspects of this study, as listed in Table 1, and were imported into the EDGAR platform as references. The selection of genomes depended on the specific question and type of analysis. The following genomes were used for most analyses: MAP-S/type III strains JIII-386 (new, finished), and S397 (draft); MAP-S/type I strain Telford; MAP-C strains K-10, JII-1961, MAP4, E1, and strain MAP/TANUSVAS/TN/India/2008 (TN/India/2008).

*M. avium* strain 104 (MAH 104), and *M. avium* strain RCAD0278 were used as references for MAH and MAA. The latter strain originated from a domestic Peking duck in China with avian tuberculosis and it was listed only as “*M. avium*” in Genbank. *M. avium* strain RCAD0278 was compared to the MAA reference strain DSM 44,156/ATCC 25291T (NZ_CP046507.1) by means of GGDC 2.1 [62,63]. *M. avium* strain RCAD0278 was reliably classified as a MAA strain by all employed analysis methods.

Draft genomes JQ5 and JQ6 which belonged to a third MAP-S sub-group were excluded from the current analyses because of insufficient quality.

## 3. Results and Discussion

### 3.1. Genome Sequencing, Assembly, and Finishing of Strain JIII-386

To facilitate comprehensive genomic comparisons with other MAP genomes, the genome of MAP-S/type III strain JIII-386 was completely closed using novel and relatively inexpensive Oxford Nanopore technology. The single, circular contig obtained from sequencing was polished with raw data from the whole-genome shotgun Illumina paired-end sequencing that was described in [34] using Pilon. These data consisted of 28.6 million 101 bp paired-end reads (~1100-fold genome coverage) and 10.9 million 100 bp mate pairs (~440-fold genome coverage). In total, 757 SNPs, 26 ambiguous bases, 18,850 small insertions, and 1075 small deletions were identified and corrected by this polishing approach.

### 3.2. General Genomic Features of JIII-386

The completed and polished chromosomal sequence of JIII-386 is 4,889,107 bp long (Figure 1). The average G + C content is 69.24%. Genome annotation with PGAP predicted 4578 protein coding sequences (CDSs), 47 tRNAs (using tRNAscan-SE 2.0; 58 tRNAs using Aragorn), three rRNAs (5S, 16S and 23S) and 24 additional ncRNAs and Riboswitches including one Rnase P, one tmRNA, and one bacterial small SRP (Appendix A). About 76% of the protein coding genes had homologs to genes encoding mycobacterial proteins with predicted function; the remaining 24% of the genome sequence consisted of homologues to hypothetical proteins or contained domains of unknown function.

Table 2 compares genomic features of the closed JIII-386 (MAP-S/type III) genome to the draft genomes JIII-386 (old), S397, and to the complete genomes Telford (MAP-S/type I) and MAP K-10 (MAP-C). The closed version of JIII-386 is about 40 kb larger, has a slightly higher GC content and includes 67 more protein-coding genes (CDS) than the draft. However, 100 more predicted pseudogenes were identified. In total, 53 genes with predicted functions and 52 genes encoding for hypothetical proteins were uniquely identified in the closed genome (Appendix A). Several other genes from the draft genome disappeared in the closed genome when annotated by PGAP rather than BacProt based on Proteinortho [68]. Because different versions of RNA annotation programs were used and also because data banks are constantly evolving, results for ncRNA detection in JIII-386 here are also different in some ways to those in the draft genome of JIII-386 [34]. Identical types and numbers of all RNAs and Riboswitches were identified in JIII-386, Telford and K-10. The closed JIII-386 genome is 18 kb smaller than Telford, but 75 kb larger than S397 and 59 kb larger than MAP K-10. The GC content in JIII-386 is identical to that of Telford (69.24%), but slightly lower than the GC content in S397 and K-10 (69.31% and 69.30%). Furthermore, a slightly lower percentage (91.95%) of the entire new JIII-386 genome contained annotated genes with predicted functions compared with that in strain Telford (92.32%), and in K-10 (92.22%).

Figure 1 shows the circular map of JIII-386 including GC skew, GC content and predicted protein-coding sequences (CDSs). In general, in genomes of many bacterial species, including *Mycobacterium tuberculosis*, the GC skew has a typical upstream-downstream structure starting at the origin of replication [69,70]. However, this structure is not visible in the MAP genomes characterized here. The current GC skew of strain JIII-386 shows a mosaic structure, which was also clearly identified by sequence comparisons with different MAP genomes represented in the multiple genome alignment Mauve plots, probably reflecting the numerous genomic rearrangements that have occurred in MAP.

### 3.3. Shine–Delgarno Motif

The Shine–Delgarno motif is part of the ribosomal binding site on mRNA genes of prokaryotes and has an important recognition function for the start of translation during the initial phase of protein synthesis. Using a suitable hidden Markov model (HMM), the Shine–Dalgarno sequence motif 5′-AGCTGG-3′ was identified in MAP JIII-386 (new), Telford, TN/India/2008, and MAA RCAD0278 genomes. This result confirms the difference from the standard pattern (5′-AGGAGG-3′) originally found by Shine and Dalgarno [71], which was described in an earlier study for JIII-386 (old), S397, K-10, and MAH 104 [34]. Figure 2 illustrates newly extracted representative Shine–Dalgarno sequences for the indicated MAP strains.

### 3.4. Chromosomal Organization

Large-scale differences in chromosomal structure of MAP genomes are illustrated in the multiple genome alignment Mauve plots in Figure 3 and Figure 4. The upper part of Figure 3 presents a Mauve alignment of the new (complete) and the old version of JIII-386 genome. Scaffolds for the JIII-386 draft genome are numbered in descending size order. Potential technical limitations of determining large-scale chromosomal structure from a draft genome were eliminated in the closed version of JIII-386. The new version provides a more useful MAP-S/type III reference genome for comparative analyses with other MAP strains. Our Mauve alignment revealed a misalignment in Scaffold 2 (old JIII-386), corrected now at position ~650 to 1900 kb in the new JIII-386 genome, and an order reversal in Scaffold 3 (old JIII-386) at position 2.5–3.4 Mb. In addition, the first region in Scaffold 3 is in reverse complementary orientation to this region in the new JIII-386 genome. The gaps between scaffolds of the draft genome sequence included in most cases larger repetitive elements, but one gap showed a higher GC content than the average. This gap could be an example of how problematic structure can influence Illumina sequencing, since this region is predicted to have a stable secondary structure.

Mauve analysis identified a similar synteny between the two MAP-S/type III genome sequences, the closed JIII-386 genome and the S397 contigs (Figure 3). This result has to be handled with reservation because of the high number of scaffolds in the draft genome of S397. An exception was one large region in JIII-386 (between 1.4 and 1.5 Mb), which was missing in the S397 draft genome. This region contains mainly genes encoding hypothetical proteins. Furthermore, there were two large putative rearrangements detected in S397 at 350 kb as well as between 1.4 and 1.5 Mb in the plot (contigs AFIF01000045.1 and AFIF01000121.1 of S397). In addition [35], a strong synteny was found among the five included MAP-C strains of different origin: K-10, JII-1961, MAP4, E1, and TN/India/2008 (Appendix A), where only one large and two very small rearrangements were identified. These results demonstrate a definite but small amount of genome flexibility among members of MAP-S/type III group and even less flexibility among members of the MAP-C group. Furthermore, the one large rearrangement detected in MAP-C genomes was previously proposed to result from a mis-assembly in K-10 [72,73].

In contrast, the structural comparative alignments among JIII-386, Telford and K-10 genomes clearly reveal the mosaic nature of MAP genomes belonging to the different sub-types MAP-S/type III and I, and MAP-C/type II (Figure 3 and Figure 4). There are many large regions with inverse orientation, rearrangements as well as small insertions and deletions. Among the MAP-S type genomes there are less gross structural differences than between MAP-S and MAP-C type genomes: Only ~35% of the total genome size was reverse complementary orientated in Telford, but ~70% differed when MAP K-10 was compared to JIII-386 (Figure 3 and Figure 4). In addition, there is also a higher block order homology among MAP-S strains than between individual MAP-S strains and K-10. Furthermore, pairwise Mauve analysis of JIII-386, Telford and K-10 (Figure 4) indicate smaller structural differences when JIII-386 rather than Telford was compared to the K-10 genome. Perhaps this could reflect a closer structural relationship between these MAP-S/type III and MAP-C strains than between MAP-S/type I and MAP-C strains. Only in very rare cases were the inverse orientated or rearranged genome regions flanked by IS elements suggesting perhaps that other mechanisms exist for catalyzing genomic rearrangements in MAP.

The mosaic nature of MAP based on MAP-S/type III compared with MAP-C genomes but without MAP-S/type I has been described previously [34], and is especially evident in non-MAP members analyzed elsewhere [35,74]. In contrast to the mosaic structure of different MAP type and subtype genomes presented here, when we compared C type MAP, the same strong synteny of genome structure was observed among MAP-C type strains as previously published [34,35].

### 3.5. SNPs and INDELs

Single nucleotide polymorphisms (SNPs) including substitutions of one nucleotide with another or an insertion or deletion of a single nucleotide are important sources of genetic variability [17]. Both SNPs in protein coding sequences and non-protein coding DNA may have functional consequences, based on non-synonymous (ns) changes (result in amino acid changes), synonymous (s) changes (do not directly influence protein sequence) or frameshifts in the encoded protein (alter the length of coding sequences), or by affecting regulatory elements. Using global sequence alignments produced with MAUVE [53], SNPs and INDELs (insertions or deletions of 1 bp–31 kb) were identified and compared between the closed genome sequences of Telford, K-10, and JIII-386 of the three main MAP types I to III. The comparisons of JIII-386 versus Telford resulted in 2613 SNPs (847 nsSNPs + 563 sSNPs (dN/dS = 1.5) + 1203 intergenic SNPs); JIII-386 versus K-10 showed 4895 SNPs (1759 nsSNPs + 841 sSNPs (dN/dS = 2.1) + 2295 intergenic SNPs), and Telford versus K-10 resulted in 4349 SNPs (1566 nsSNPs + 763 sSNPs (dN/dS = 2.1) + 2020 intergenic SNPs). In addition, for JIII-386 compared with draft genome S397, 1692 SNPs (589 nsSNPs + 302 sSNPs (dN/dS = 2.0) + 801 intergenic SNPs) were determined.

Detection of half as many total SNPs among MAP-S strains relative to comparisons between MAP-S and MAP-C strains confirm the closer relationship of the MAP-S (type I and III) strains than either to K-10, as expected. Furthermore, JIII-386 and S397 (both type III) show the closest relation with about 1700 SNPs in total and this close relationship is especially evident from the low number of intergenic SNPs. However, the finding of 2613 SNPs reveal a significant diversity between JIII-386 and Telford affirming that they are members of different subgroups.

The general higher number of non-synonymous relative to synonymous SNPs (dN/dS ratio) in MAP was discussed before as an indication for ongoing adaptive evolutionary processes possibly to different hosts or environments acting on protein coding genes, and likely reflects the relatively short evolutionary age of MAP [75,76]. Furthermore, a substantially lower dN/dS ratio was identified in 14 MAH and two MAA genomes and was suggested to be associated with a stabilizing selective pressure in these strains [35]. Obligate *Mycobacterium tuberculosis* complex pathogens also have a higher dN/dS ratio [77]. It will be important to examine more genomes of obligate pathogenic strains of MAA to determine if high dN/dS is a MAP specific phenomenon in MAC.

Table 3 shows the number of insertions detected in each strain in comparison to others and their total length. Notably, insertions of single nucleotides are also included. Altogether, this means that MAP-S genomes JIII-386 and Telford, share 674 INDELs comprising only 33,314 bp. In contrast, MAP-S and MAP-C genomes are more distinct; Telford and K-10 share 716 INDELs with 205,458 bp, and JIII-386 and K-10 share 1004 INDELs with 262,650 bp. Results shown in Table 3 are also reflected in the differences between the total sizes of these genomes (Table 2).

### 3.6. Taxonomic Calculations

The relationship between these closed genomes was determined using GGDC 2.1 with formula 2 [62,63], based on G+C content differences and comparison with DNA–DNA hybridization (DDH) similarities (calculated in silico). This calculation resulted in 99.40% similarity for JIII-386 and Telford, 98.60% similarity for JIII-386 and K-10, and 98.70% similarity for Telford & K-10. This method definition is based on genomic evolution in other bacteria that tend to evolve more rapidly because of their higher tendency to take up and incorporate foreign DNA. In this method definition, >70% means identical species and >79% means identical subspecies. Although it is known that MAP is a monomorphic linearly evolving subspecies, the differences among the three MAP type strains are surprisingly far below the subspecies level. However, the data still show the closer relation between the both MAP-S genomes than between the MAP-S and MAP-C genomes.

### 3.7. Specific Genomic Elements

#### 3.7.1. CRISPR

Clustered regularly interspaced short palindromic repeat (CRISPR)/CRISPR-associated protein (Cas) systems are prokaryotic adaptive immune systems that protect against invading nucleic acids such as phages or plasmids. The presence of a CRISPR-Cas defense system was tested using the CRISPRCasFinder [78]. Only short repeats without closely located *cas* genes were found in JIII-386 and the other analyzed MAP genomes, and the number detected in each genome that was analyzed in this manner is indicated in parentheses in Table 4. The CRISPRCasFinder classified these repeats as “questionable” representing regions that include potential CRISPR repeats, but the hits were under the threshold for a functional CRISPR-Cas region. This same pattern was also detected in the genomes of Telford, S397, and MAP K-10. These observations are in accordance with the findings of He et al. [79] who performed a comparative analysis of CRISPR-Cas regions in different *Mycobacterium* strains, and found that larger CRISPR repeats in close proximity to *cas* operons were detected in *M. tuberculosis,* but only short repeats and no nearby *cas* genes were detected in MAH 104 and MAP K-10. Results suggest that the detected conserved short repeat sequences in the newly sequenced MAP-S strains are relics of similar now defunct CRISPR-Cas defense systems in MAP, or they are perhaps active components of an undefined alternative defense system.

#### 3.7.2. Prophages and Prophage-Like Elements

Prophages are important genetic components that are transferred horizontally and can impart bacterial genome variability, evolution, and virulence properties [80,81]. Some of them contribute to the adaptation of bacteria to specific ecological niches [82]. In this study, both recent MAP-S type genomes, JIII-386, and Telford, were examined for prophages and prophage-like elements. The PHASTER prophage finder software identified one 17.7 kb region in JIII-386 and in Telford that includes a potential fragment of a prophage genome, but with a low score (incomplete). Appendix A shows the length and position of this region within JIII-386 and Telford; nine genes were related to prophage genes, but to different reference phages (Appendix A). In contrast, using PHASTER software on the MAP K-10, there were no prophage or prophage-like element identified, confirming results of Fan et al. for K-10 performed using other software [83]. However, manual inspection of the K-10 genome for the phage related genes in JIII-386 using Geneious uncovered a similar region in K-10. There were no detected frameshifts of these genes in K-10; thus, the reason that they are not detected by PHASTER is unclear-presumably too low a score for prophage sequences identities in this MAP-C genome. Appendix A lists the predicted CDSs of this potential prophage fragment in JIII-386, and the homologs in Telford and K-10, along with their different suggested reference phages. This predicted region of a prophage fragment corresponds with an individual genomic island, which was identified by Island Viewer in all included MAP-S and MAP-C strains (see further below). In addition, in several other determined genomic islands of these MAP strains individual prophage-like genes were found by manual inspection together with many short genes annotated as “hypothetical”, which were all orientated in the same direction and either upstream, or downstream frames. Two of these clusters in each strain correspond with putative prophages previously identified by Alexander et al. in K-10 [38]; others may represent new unknown complete or incomplete prophages fragments. There was no difference in the number of prophage-like-elements determined between MAP-S and MAP-C type strains.

#### 3.7.3. Insertion Sequences

Insertion sequence (IS) elements are a type of transposable element which play an important role in creating genomic diversity and plasticity of bacteria, enabling them to adapt to new environmental challenges and or to colonize new niches [84]. Based on closed genomes, for JIII-386, Telford, and MAP K-10, altogether twelve different IS elements belonging to six families (IS*110*, IS*256*, IS*481*, IS*1182*, IS*30*, and IS*3*) were identified by ISfinder software and by using annotation results and sequence comparisons. The total number of these transposable elements differed between the investigated strains (Table 4). The detailed data for this analysis are presented in Appendix A. The most frequently detected transposase CDS was IS*900* (IS*110* family), followed by IS*1311* (IS*256* family). The Telford genome (MAP-S/type I) exhibited more copies of IS*900* (n = 22) than JIII-386 (n = 18) and K-10 (n = 17). A recently published analysis has shown that reliable assembled and closed MAP-C genomes exhibited 17 or 16 copies of IS*900* [35]. However, genes upstream and downstream of the additional IS*900* elements in Telford are also present in the other strains, but without the IS*900* element suggesting either that the IS element inserted additional copies of itself into new locations in the Telford genome or that these elements were excised from these regions in the other genomes. Both MAP-S genomes had a higher number of IS*1311* (n = 9) elements in comparison to Map K-10 (n = 7). Only one IS3 family transposase was found in K-10, but there were three in JIII-386 and four in Telford. Furthermore, there were six hits (K-10) and seven hits (JIII-386, Telford) against an IS*1182* family transposon known as ISMap02 [85]. The sequence identity of this element was checked by sequence comparisons [86]. All results are shown in detail in Appendix A. The higher number of IS elements in MAP-S than in MAP-C genomes and in particular in the type I Telford genome could be associated with the higher diversity within MAP-S type subgroup but more closed Type 1 genomes will need to be studied before any definitive conclusions about evolutionary mechanisms can be proposed.

### 3.8. Putative Virulence Associated Genes and Gene Clusters

Marri et al. [87] compared five genome sequences of four different mycobacterial species (*M. tuberculosis*, *Mycobacterium bovis*, *Mycobacterium leprae*, and MAP K-10) and uncovered many genes that were conserved encoding for key metabolic pathways such as energy metabolism, amino acid biosynthesis, cofactor biosynthesis, nucleotide metabolism, and macromolecule metabolism. The major differences among these species were detected gene products involved in survival under diverse anaerobic, microaerophilic or aerobic conditions inside or outside the host, products for cell wall maintenance and gene families encoding the acidic glycine-rich proline-glutamate/proline-proline-glutamate (PE/PPE/PGRS) proteins. However, only genes belonging to the Mammalian cell entry (mce) operons, to the PE/PPE motif, to metal uptake, to some secretions systems or secreted proteins, to stress adaptation, and to cell envelope synthesis genes were designated as putative virulence associated genes in MAP. To reveal differences among the MAP-C and MAP-S/type I and III, the content of some of these genes and gene clusters was compared between JIII-386, Telford and K-10 (Table 4, Appendix A).

#### 3.8.1. Mce Genes

*Mce* genes were first discovered in *M. tuberculosis* and were associated with mycobacterial cell entry (mce) in non-phagocytic mammalian cells [88]. They were identified in both pathogenic and non-pathogenic environmental mycobacteria as well as in many other bacterial species. In mycobacteria, these genes exist as operons encoding putative membrane associated proteins [89]. There are four *mce* operons in *M. tuberculosis* (*mce1* to *mce4*), each with two *yrb*E genes and six *mce* genes [90]. Additional *mce* operons were detected in members of the *Mycobacterium avium* complex (MAC). There are eight *mce* gene clusters in MAP [89]. Individual clusters of *mce* genes in different mycobacterial genomes could possibly encode specific control mechanisms for individual adaptations that have evolved to contribute towards entry and survival in diverse hosts and environments [91]. Furthermore, the expression of *mce* operons could be modulated in response to environmental signals, such as nutritional status and stress conditions [92].

In accordance with Casali and Riley [89], the current analysis uncovered eight separate *mce* clusters, containing six *mce* genes in JIII-386, Telford, and MAP K-10 genomes (see Appendix A) and in six of these clusters a pair of upstream genes encoding ABC transporter permease (NCBI annotation) genes. These genes show different lengths and sequences (identity < 66%) in the different *mce* operons. In *M. tuberculosis,* there are the two copies of the gene *yrb*E gene: *yrb*EA and *yrb*EB [91]. One *mce* operon in MAP (mce8, Appendix A) harbors one rather than two ABC transporter permease CDS upstream of the *mce* genes. In addition, upstream of *mce* operon mce6 (Appendix A) in all three strains there are no ABC transporter permease CDSs and instead three other genes were identified, pointing in the reverse direction, starting with cyclohexane carboxylate-CoA ligase CDS in JIII-386, with AMP dependent synthetase CDS in Telford, and with AMP-binding protein CDS in K-10. Of particular interest is the absence of homologous genes for *mce*2C, *mce*2D, *mce*2E, and *mce*2F within the mce2 gene cluster in JIII-386. These deletions and the deletion of two other genes, including *rec*D, were described as region LSP^A^11 and are absent in some MAP-S type strains [38]. Here we determined that JIII-386 but not Telford belongs to this subgroup. The influence of these deleted genes on the virulence of the different MAP sub-types is worthy of further investigation. Furthermore, one difference between MAP-S and MAP-C genomes concerning *mce* genes could be confirmed: There were additional individual *mce* CDSs identified outside of detectable *mce* operons; these include two adjacent genes in JIII-386 (CDQ89_02215 and CDQ_02220) and in Telford (EGA31_02785 and EGA31_02790), for which there is only a single CDS in K-10 (MAP_RS16900 (MAP3289c)). These genes are homologues to *mce* genes MAV_4125 and MAV_4126 in *M. avium* strain 104 as described by Castellanos et al. ([39] 2009). These slight differences in *mce* clusters may be important for the development of different phenotypic characteristics in different MAP types.

#### 3.8.2. Membrane Protein and Transporter Genes

Membrane proteins and membrane protein transporters constitute important components of mycobacterial cell walls and cell envelopes [93]. Mycobacterial membrane protein Large (MmpL) transporters represent a subclass of RND (Resistance-Nodulation-Division) transporters, which participate in the export of lipid components across the cell envelope. These surface-exposed lipids with unusual structures could play key roles in the physiology of mycobacteria and/or act as virulence factors and in immunomodulation [94]. The MmpLs have differing substrate specificity and mechanisms of regulation. Finally, variations in the cell envelope might lead to major differences in the virulence characteristics of individual mycobacterial subspecies or subtypes [87]. Differences in presence or absence of individual *mmpL* genes and their products in MAP cattle or sheep type strains could help in determining host specificity [95].

Altogether 16 sequence loci were identified in MAP-S (JIII-386 and Telford), and 17 sequence loci in MAP-C (K-10) genomes, which were annotated as MmpL or RND family transporter CDSs (Appendix A). Gene comparisons in Geneious show that all these genes and their neighboring genes are present in all three genomes. *MmpL* and *rnd* genes have pairwise sequence identities ranging between 99.8 and 100%. In JIII-386, at one sequence locus (homolog to MAP_RS13435 and EGA31_06370) two genes are annotated (CDQ89_06300 and CDQ89_06305), and both are predicted to be frameshifted. Furthermore, in all three searched MAP genomes some *mmpL*/*RND* genes were annotated as pseudogenes (based on frameshifts). In JIII-386, this applies to four out of 10 *mmpL* gene loci. Homologous *mmpL* genes in MAP-S/type III strain S397 also show frameshifts. In Telford one *mmpL* and two *rnd* genes which were not homologous to genes in type III strains, and in K-10 one *rnd* gene (homolog to the also affected *rnd* gene EGA31_15435 in Telford) were classed as pseudogenes. Pseudogenes are found mostly in bacteria that are obligate parasites. These genes lose their function under the selective pressure or because of adaptation to specific host environments. The differences detected within the *mmpL* gene group represent additional examples of genome flexibility during the evolution of the different MAP types.

In addition, only MAP-C type strain K-10 has one specific MmpL family transporter CDS (MAP_RS08845 (MAP1738)), which exhibits 76% sequence identity with the *mmpL*5 in *M. tuberculosis* H37Rv (NC_000962). The presence of a homologous gene to *mmpL*5 in MAP cattle type strains and its absence in sheep type strains because of a deletion including several genes was described before [95] and was associated with differences in culture requirements and host specificity of different MAP types.

#### 3.8.3. PE/PPE/PGRS Genes

This multigene family encodes acidic glycine-rich proteins, which are secreted or localized to the cell surface and which are unique to mycobacteria. These proteins are thought to influence antigenic diversity, virulence, host adaptation and the growth in macrophages [96,97]. For unknown reasons, genes of this family are less abundant in MAP genomes (1%) than in *M. tuberculosis*, where they comprise approximately 10% of the proteome [98]. Most PE and PPE sequences are conserved among the *M. avium* species [96]. In the current analysis, the hidden Markov model [99] used for the detection of PPE-/PE-/and PE-PGRS genes in MAP-S/type I and III, and MAP-C revealed a high conformity in the total number of PE/PPE/PE_PGRS family protein CDS.

The MAP strains compared here have 43 PE- and PPE family CDSs (see Table 4). Homologous genes in Telford and K-10 to PPE genes in JIII-386 show average nucleotide sequence identities of 99.6 and 99.8%, respectively. One PPE gene of K-10 (MAP_RS08825 [MAP1734]) is included in gene region of deletion #2 which is absent in MAP-S strains. This region also contains the above-mentioned MAP-C specific *mmpL* gene (MAP_RS08845). In MAP-S strains another unique PPE gene is included within the MAP-S specific region LSP*^S^* I. The single Pro-Glu-polymorphic GC-rich sequences (PE-PGRS) gene identified in both MAP-S strains JIII-386 and Telford is truncated and therefore probably does not function; in K-10 one PE-PGRS homolog sequence is annotated as hypothetical protein. Furthermore, the total number of PE/PPE family CDS in MAP is also not very different from that in other members of the species *M. avium*: MAH strain104 (6 PE + 36 PPE) and the obligate pathogen MAA RCAD0278 (6 PE + 33 PPE) suggesting that this type of function is slow to evolve in MAC.

#### 3.8.4. Mycobactin Gene Cluster

Mycobactin is a siderophore responsible for the binding and transport of iron into the bacterial cells, which is essential for electron transport and the function of many metabolic processes. A cluster of 10 genes (*mbtA-mbtJ*) was described in *M. tuberculosis* encoding mycobactin which aides with the transport of iron [100]. Homologs to this cluster were also identified in MAP and other members of the MAC complex. Significant differences were detected in the primary structure of this region of MAP K-10 and S397 (MAP-S/type III) relative to *M. tuberculosis* [28,101]. In contrast to laboratory cultivation of MAH or MAA strains, cultivation of MAP requires supplementation with mycobactin because of the presumed inability of MAP to produce this important protein under in vitro conditions. However, the presumed limiting factor in mycobactin production in MAP, a truncated key (salicyl-AMP ligase) gene *mbtA* in K-10 [28], could not be confirmed, because *mbtA* is harbored in MAP S397 and is predicted to be the same length as the locus in *M. tuberculosis* [101].

Figure 5 illustrates that all ten homologous genes of the mycobactin cluster were identified in MAP-S/type I and III strains Telford, JIII-386, and S397. Relative operon gene content is compared here with K-10, MAH 104 and MAA RCAD0278 (see also Appendix A). Among the studied genome sequences, the structure of the mycobactin cluster including genes *mbtA* to *mbtH* is highly conserved. However, major differences were found for *mbtA* and *mbtE*, and in the gaps between specific genes. The *mbtA* gene is shorter in K-10 (401 aa) than in JIII-386 (547 aa), Telford (552 aa), MAH104 (552 aa), and MAA RCAD0278 (546 aa). A ~200 bp region within *mbtA* differs between K-10 and the both MAP-S strains (with only 37% identity). The *mbtE* gene is longer in K-10 (plus 50 aa), MAH 104 (plus 61 aa), and RCAD0278 (plus 62 aa) when these loci are compared to the one in JIII-386 and Telford (Figure 5). Downstream, this gene shows only 40% identity at the nucleotide level when a 2.7 kb sequence of *mbtE* in JIII-386 (CDQ89_13975) and Telford (EGA31_07100) is compared to this same region in K-10 and also to MAH 104 and MAA RCAD0278. The other part of the *mbtE* gene in K-10, shows a very high identity (99.8%) to homologous regions in the MAP-S strains. MAH 104 and MAA RCAD0278, have an *mbtE* pairwise identity of 86.8%. Remarkably, several genes are frameshifted: *mbtE* and *mbtF* in K-10; *mbtA*, *mbtB*, *mbtE*, and *mbtF* in JIII-386; and *mbtD* and *mbtE* in Telford. In addition, *mbtC* is interrupted in JIII-386 and Telford (see Appendix A). Furthermore, in all these strains, there are gaps downstream of the mycobactin cluster between *mbtA*, *mbtJ*, and *mbtI*. Between genes *mbtA* and *mbtJ*, in K-10 there is a 19.4 kb gap, and in both MAP-S strains JIII-386 and Telford there is a 53.3 kb gap, in accordance with data reported for K-10 and MAP-S strain S397 [101].

In the MAP genomes compared here, this gap includes one *mce* gene cluster (mce6), which is MAP specific [36]. Mce genes are missing in this region of MAH 104; however, in MAA RCAD0278 this region contains two mce clusters. Furthermore, between genes *mbtJ* and *mbtI* there is a 7.2 kb gap in both K-10 and JIII-386. This result in the newly closed JIII-386 differs from that for MAP-S/type III strain S397 of Timms et al. [101], who did not find a gap between these genes. In Telford the mycobactin cluster is in inverse orientation and the gene *mbtI* is situated far away (1.83 Mb to *mbtA*) but in the same relative orientation as the other genes in the operon. It is not clear at this point if this difference is typical of MAP-S/type I strains, is specific to Telford and its close relatives or is an assembly artefact. This will become clear as additional finished type S1 genome sequences become available for comparison.

In summary, the numerous differences detected among the mycobactin gene clusters, present in the genomes of different MAP type strains (sequence differences in *mbt*A, *mbt*E, interruptions in *mbt*C in MAP-S strains, different gap lengths between *mbt*A and *mbt*J), MAH and MAA strains compared here are likely to account for some of the differences in their growth characteristics including their different dependencies on Mycobactin J for the in vitro growth.

The significance of each of the sequence differences detected in putative virulence-associated genes are as yet unclear. Most of these genes are present in similar numbers in the different MAP sub-types compared here, suggesting that pathogenic characteristics such as host association are likely to depend in part on differences in transcription and or translation of important gene sequences and perhaps on genes other than those described above.

### 3.9. Secondary Metabolite and Antibiotic Resistance Genes Clusters

#### 3.9.1. Secondary Metabolite Genes

Secondary metabolite and antibiotic resistance gene clusters were compared to better understand genome flexibility within the studied MAP subtypes. The antiSMASH 5.1.2 platform which incorporates information from a variety of different sources to identify and annotate secondary metabolite biosynthesis gene clusters in bacterial and fungal genomes [58], was employed to identify biosynthetic loci of secondary metabolite compound classes in the three MAP subtypes. A total of 15 secondary metabolite clusters were identified in JIII-386 and in Telford and 16 were detected in K-10. When the genomes of the three subtypes were compared, four clusters encode polyketide synthases (PKS), four (JIII-386 and Telford), and five (K-10) encode non-ribosomal polypeptide synthases (NRPS), two (JIII-386, K-10), and three (Telford) encoded mixed NRPS-PKS clusters, one terpene synthase, one encoded a mixed PKS (only in JIII-386 and K-10), and one other is designated as a bacteriocin cluster.

In JIII-386, two clusters with the predicted metabolite synthesis functions for mycobactin and alkylresorcinol were identified with 60% and 100% gene similarity to the antiSMASH database (Table 5). Homologous clusters were identified in the Telford with 70% and 100% similarity and in K-10 with 90% and 100% identity to the database. Remarkably, K-10 showed an additional cluster with 100% similarity (predicted product: methylated alkyl-resorcinol/methylated acyl-phloroglucinol). Furthermore, with exception of the “Terpene metabolite cluster” (predicted product: isorenieratene) with a similarity of 45% and 57% to clusters in the antiSMASH database for all three genome sequences, the other metabolite clusters showed a very low similarity (6–35%), so their role in the lifestyle of MAP is unclear.

Altogether, MAP-S strains JIII-386 and Telford share 14 of 15 clusters. The PKS cluster predicted to synthesize marinacarboline is only found in JIII-386 and also in K-10, but not in Telford, whereas the Telford and the K-10 genomes harbor an additional NRPS cluster with a prediction for chloromyxamide synthese activity. MAP-S genomes include a “NRPS-PKS cluster” with prediction for salinamide synthesis and a “PKS cluster” with prediction for lagunapyrone A-C, both not detected in MAP-C type strain K-10. In K-10, two clusters are unique (i) a putative “methylated alkyl-resorcinol/methylated acyl-phloroglucinol cluster” with 100% similarity to clusters in the antiSMASH database, and (ii) an undefined glycopeptidolipid cluster which showed a very low similarity to clusters in the antiSMASH database.

#### 3.9.2. Antibiotic Resistance (AR) Genes

MAP is a member of Nontuberculous Mycobacteria (NTM), who have both natural and acquired types of resistance mechanisms towards several antibiotics and some of these are very similar to those of *M. tuberculosis* [102]. Generally, antibiotic drugs are not used for treatment of paratuberculosis infected animals so there is less selective pressure on these gene targets. Two mechanisms are thought to affect the natural drug resistance of mycobacteria, and are probably also involved in the drug resistance of MAP: (I) the very low permeability of mycobacterial cell envelope [103] and (II) different active multidrug efflux pumps [104]. Furthermore, the primary mechanism for driving drug resistance of mycobacteria is the acquisition of spontaneous mutations, SNPs, INDELs, or sometimes-large deletions [105] in genes that code for drug targets or drug-activating enzymes rather than via horizontal gene transfer of resistance genes on mobile genetic elements as is common in many other bacteria.

Nevertheless, ARG-ANNOT software [59], which probes genomes for homology to known AR genes, was used to search for potential antibiotic resistance genes. As expected, no antibiotic resistance genes were identified in the JIII-386 genome. The best hit against the ARG-ANNOT database represents a transporter in JIII-386 that shows 70% sequence identity and 98% query coverage to a known ABC-type multidrug transport system. Homologous loci were found in all of the different *M. avium* strains included in this study (Telford, K-10, S397, MAH 104, and MAA strain RCAD0278). ABC transporters were demonstrated to be involved in drug resistance in *M. tuberculosis* [104]. In the current study, the best hit to typical clinical antibiotic resistance genes, e.g., *bla*-genes, showed at best less than 20% sequence identity leading to the conclusion that JIII-386 does not harbor any of these typical antibiotic resistance genes. This was also true when the other studied MAP strains (S397, Telford and K-10) were screened with ARG-ANNOT.

SNPs in a specific 81-bp region of the *rpoB* gene are associated with Rifampicin (RIF) resistance in *M. tuberculosis* [106]. Beckler et al. [107] determined that several MAP isolates that were resistant to RIF and to Rifabutin, also had mutations in the *rpo*B gene (compared with K10). In the current study, the complete *rpo*B gene sequence was found to be identical in JIII-386, Telford and K-10. Mutations in genes contributing to specific antibiotic resistance in members of *M. avium* is rare and some negative results were discussed with reference to other mechanisms [108]. With few detected sequence differences in likely antibiotic resistance gene targets, it is not possible to draw conclusions about the antibiotic resistance of MAP strains without in vitro studies.

### 3.10. Genomic Islands

Genomic islands (GIs) are specific parts of bacterial genomes containing clusters of genes that are likely to have been introduced relatively recently by horizontal gene transfer. The genes in GIs are often associated with microbial adaptation; they encode for a variety of functions including pathogenicity and antibiotic resistance and have had a substantial impact on bacterial evolution [109]. GIs are characterized and identified by differences in sequence composition such as GC percentage, codon bias and by a higher content of transposases and integrases in comparison to the rest of the genome sequence. For better understanding MAC evolution, Island Viewer 4 software was employed to uncover additional insertions and deletions of GIs and their encoded gene functions in the finished genomes of different MAP subtypes, MAH, and MAA.

Figure 6 illustrates detected GIs in circular chromosomal maps of JIII-386, Telford, K10 and TN/India/2008 MAP genomes, as well as the genomes of MAH 104 and MAA RCAD0278. The virtual chromosomal location of some of these GIs are identical, whereas others are different probably because of the mosaic nature of MAP genomes. This mosaicism is additionally demonstrated in the circular maps of MAP strains by the chromosomal location of ten corresponding GIs among these genomes that are labelled by numbers 1 to 10 in the plots. These plots and Appendix A show that MAP-S type I strain Telford has the highest number of GIs (n = 22) in comparison to between 16 to 18 GIs in JIII-386 and the MAP-C strains K-10, TN/India/2008, and JII-1961. This could be a hint that MAP-S/type I strains are slightly phylogenetically older than MAP-S/type III and MAP-C strains. The identified GIs comprise 4 to 28 CDS (6 to 26 kb), altogether including about 180–270 CDS (two thirds with predicted function and one third encoding for hypothetical proteins) with a total size of about 200–260 kB. Corresponding GIs that were conserved in MAP-S/type I, III, and MAP-C strains showed differences in the total sizes and orientation within the genome sequence as well as in the number of detected genes. Some genes were frameshifted in individual strains.

Even when considering only unique CDSs, there were no MAP-type or strain specific GIs identified. Most CDSs of additional GIs detected in individual or multiple strains were present in all searched MAP genomes, but were outside of regions defined as GIs by Island viewer. An exception were 12 genes that were detected in the genomic islands J10 in JIII-386 and T8 in Telford (Appendix A) but are completely absent in MAP-C strain genomes. These genes belong to the MAP-S specific LSP^S^ I region which will be discussed in more detail below. Furthermore, individual GIs were partly homologous to MAP-associated and MAP specific LSP^P^s (LSP^P^ 1, 8, 13; and LSP^P^ 4, 11, 14, and 16), including highly conserved genes [36]. These genes were absent in MAH 104. Most genes in the other GIs were also identified in MAH 104.

The GC content of the GIs ranged from 59.4% to 74% in comparison to the average GC content of about 69% in the rest of the genome. There is about a sixfold higher frequency of putative transposases within the identified GIs relative to the rest of the genomes. In addition, in each strain three or four GIs contained genes encoding putative prophage integrases or other putative phage-proteins. These GIs are homologous among the MAP strains. One of these GIs corresponds with the prophage fragment region identified by PHASTER (GIs: J6 in JIII-386, T7 in Telford, K12 in K-10, JII-12 in JII-1961, and TN10 in TN/India/2008; Appendix A); two others correspond with LSPP4 and LSP*^P^*11, described as putative prophages by Alexander et al. [38].

As expected, several genes in the identified GIs are predicted to have an impact on the virulence of MAP. For example, four to six genes encode putative PE/PPE family proteins distributed within three to five GIs. In addition, some other virulence-associated genes were detected in individual GIs, for example, CDSs for WXG100 family type VII secretion systems or different family transcriptional regulators. Mycobacterial pathogens use specialized type VII secretion systems to transport crucial virulence factors across their unusually complex cell envelopes into infected host cells [110].

Furthermore, the GIs include genes that were identified by Marri et al. [111] as possibly acquired by lateral gene transfer (LGT) by the common ancestor of MAP, MAA, and MAH, very soon after the divergence of MAP. This applies in the current study to 43 genes (20%) in 11 out of 18 GIs in MAP K-10. Most of these genes (n = 26) belong to two GIs encoding putative prophage integrases and a large number of hypothetical proteins including the MAP specific gene cluster LSP^P^4. Other genes likely acquired by LGT belong to a GI homolog of LSP^P^11 (LSP^P^s definition see [38] and below).

Several large regions of difference between MAP-S and MAP-C genomes were not recognized as genomic islands. These include the MAP-S specific LSP*^S^* regions I–IV (see below); these regions were probably lost from the MAP-C lineage very recently and do not have features that are typical of GIs. Exceptions are the above mentioned 12 out of 32 genes of LSP*^S^* I (CDQ89_14105-14160), which are absent in MAP-C, and 10 of these genes which are also absent in the studied MAH and MAA genomes (see below in chapter 3.12., Appendix A). These genes (CDQ89_14115-14160) belong to LSP*^S^* region Ia and show high sequence identities with genes in other species of the *Mycobacterium avium* complex (*Mycobacterium marseillense*, *Mycobacterium paraintracellulare*, *M. intracellulare*). Further analysis is required to reveal the importance of these genes in MAP-S and MAP-C evolution.

The zinc responsive genomic island (ZNGI) described by Eckelt et al. [112] in MAP K-10, also did not meet the criteria for GI as defined by the Island Viewer software. Genes of this island encode a specific metabolic pathway and are present in both MAP-S strains (JIII-386 (CDQ89_21205-21525) as well as in Telford (EGA31_21515-EGA31_21835). Their order in these genomes is identical to those in K-10, but in reverse orientation.

The distribution of GIs was compared with that in the MAC members MAA RCAD0278 and MAH 104. In MAA strain RCAD0278, only seven regions were identified as GIs, including 75 CDS (comprising 71.4 kb; Appendix A). One GI (MAA4) corresponded with a GI in MAP and two GIs (MAA2 and MAA3) corresponded with GIs in MAH 104. Individual genes of three other GIs are also present in MAP but not in regions defined by Island Viewer as genomic islands. Two islands contain putative prophage integrase or other putative phage-encoded proteins and a high number of hypothetical proteins—different from phage genes detected in MAP and MAH 104 genomes. Four out of seven GIs contain transposable elements belonging to IS*21* and IS*100* transposase families that were not present in MAP.

In contrast to the MAP and MAA, MAH strains are more ubiquitous and opportunistic members of MAC complex [5]. In strain MAH 104 a significantly higher number of GIs (n = 47) were identified than in MAP and MAA, including 563 genes comprising 546 kb. Five islands included genes associated with presumed prophage like elements, different from those in MAP genomes. Altogether the MAH GIs contained 12 PE/PPE family protein CDS, 13 *Mce* family protein CDSs, 27 TetR family transcriptional regulator CDS and 26 transposable elements; belonging to IS*3*, IS*21*, IS*256*, and IS*1380* families. These findings suggest the more frequent exchange with other types of bacteria and a higher frequency of horizontal gene transfer by MAH strains relative to the more specialized, obligatory pathogens MAP and MAA. There is little known about differences in pathogenicity of MAH in comparison to MAP and MAA in different hosts. The identification of different numbers of putative virulence genes among these *M. avium* members hint that differences are more likely to be based on complex differences in regulation of genes and other unknown mechanisms than on the pure presence of these genes.

### 3.11. Phylogenetic Analysis based on Extended Panel of MAC Genomes

The phylogenetic relationship of JIII-386 and Telford to others in the *M. avium* complex was investigated using currently available and recently published closed MAP and MAA genomes, and in addition five MAH genomes of different geographical origin. Since there were no other closed MAP-S genomes for comparison, the draft genomes of S397 and CLIJ361 were also included. All these strains are listed in Table 1. Phylogenetic analysis was carried out based on the amino acid (AA) sequences of all 2807 orthologous CDS that are conserved among these 24 different *M. avium* complex strains. The resulting phylogenetic tree in Figure 7 shows the known clustering of MAP-S types, which contained subgroups Type I and III, MAP-C (type II), and MAA. The five members of the MAH group, which generally possess the highest diversity within *M. avium* subspecies [5], are located in three distinct branches/clusters of the tree between the MAP and the MAA cluster.

The MAH strain 104 genome is most closely related to the characterized MAA genomes. The Egypt MAP strain E93 genome is located outside of the main MAP-C cluster. Additional studies are needed to clarify the apparent special features of this genome.

The current tree showed a very similar phylogeny to what was determined recently by Bannantine and colleagues [35] even though the two phylogenetic relationships were based on other input data and different tools and parameters. Five of the same MAH strains, two out of the three MAA strains, strain Telford, and all MAP-C strains (except E93) were used in both comparisons.

All genomes shown in Figure 7 except E93 and the draft genome of CLIJ361 were used in the following core and pan genome analysis.

### 3.12. Validation of MAP-S Specific LSP^S^ Regions and Deletions Based on the Core Genome Analysis

Different studies in mycobacterial genomics showed that loss or insertion of large sequence regions—large sequence polymorphisms (LSPs)—are major contributors to genetic diversity and are presumed to be important in the evolution and virulence *M. tuberculosis*- and *M. avium* complex bacteria. Numerous LSPs have been revealed by DNA microarray-based comparisons or in silico analysis in *M. avium* strains [36,38]. MAP strains exhibit a characteristic profile with a set of large genomic insertions (LSP^P^s) that are absent in other *M. avium* strains and show a characteristic deletion (LSP^A^8), [38]. Furthermore, the two major MAP sub-groups (MAP-S and MAP-C) are characterized by chromosomal gene acquisition and gene loss. Alexander et al. [38] listed three regions, which were present in different MAP-S strains but absent in MAP-C strains (LSP^A^4-II, MAV14, LSP^A^18) and two MAP-S specific deletions. More recently, Bannantine et al. [33] suggested 10 MAP-S lineage-specific LSP*^S^*s, five of which were ruled out as a means for distinguishing MAP types by additional studies [34].

Here, the de novo assembled JIII-386 genome, the recently closed Telford genome, the last available annotation of S397 (MAP-S), 12 MAP-C, five MAH, and three MAA RCAD0278 genomes were searched by core genome analysis (EDGAR 2.0) and manually for the presence or absence of MAP-S specific regions comprising LSP*^S^*s and deletions (Figure 8). The recently defined MAP-S specific regions LSP*^S^* I–IV [34] including distinct previously published regions were verified and improved by this comparative approach (Table 6). Appendix A lists all genes that are common to JIII-386, Telford, S397, MAH 104, and MAA RC0S0278 along with their predicted functions. For S397, new genes were identified that were assigned to the LSP*^S^*s, resulting in a high degree of conformity of LSP*^S^* I–IV among the three MAP-S strains. Only nine genes could not be annotated in all three MAP-S strains simultaneously in these regions. In addition, there were also several differently distributed frameshifted genes.

In this study the presence of LSP*^S^* regions I–IV was confirmed for MAP-S strains JIII-386, S397, and strain Telford; these regions were absent from all of the 12 MAP-C genomes (see Table 6 and Figure 8), but were present or partly present in several of the *M. avium* subsp. *hominissuis* and *M. avium* subsp. *avium* strains that were included in these analyses.

Genes with predicted function were classified into different metabolic pathways via PRIAM and KEGG database comparisons. Only a few genes belong to a described KEGG pathway. However, in general, the singleton gene set of the LSP*^S^*s includes different transcriptional regulators belonging to different families and two genes coding for different cytochrome P450 proteins that could influence virulence of mycobacteria [114]. In addition, genes in LSP*^S^* III comprising three methyltransferases and one desosaminyl transferase precursor are likely involved in the synthesis of a macrolide(s). Some macrolides have antibiotic or antifungal activity. Furthermore, about 14% of MAP-S specific genes were predicted as hypothetical proteins with unknown functions.

There are three gene clusters comprising about 32 genes that were annotated in MAP K-10 and were previously determined to be absent or present in MAP-S type strains: deletion s∆−1 (MAP1432-MAP1438c), deletion #1 (MAP1484c-MAP1491) and deletion #2 (MAP1728c-MAP1744), see [33,34,37,40,95,115]. Genes of all three deletions were found to be absent in sheep strains from United States including S397 [33]; genes of deletion #1 and #2 were also absent in Australian sheep strains [115]. However, genes belonging to deletion s∆−1 were found to be present MAP-S strains JIII-386 and CLIJ361 from Germany and Australia, whereas genes in deletion #1 and #2 were absent in these genomes [34]. In this study, using the new complete genome sequences of Telford and JIII-386, the presence or absence of genes of these three deletions was re-examined and analyzed in a total of 11 MAP-C, 5 MAH, and three MAA genomes (see Figure 8, Appendix A).

In contrast to strain S397, which had only one detected orthologous gene in deletion s∆−1 region (MAPs_20200 in S397; MAP1438c [MAP_RS07310] in K-10), in MAP-S strains JIII-386 and Telford seven out of eight genes of deletion s∆−1 region are present in a gene cluster (CDQ89_12125-12155; EGA31_08945-08915). The homologous gene of MAP1432 [MAP_RS07275] was found at another location. Furthermore, deletion of gene clusters #1 [MAP_RS07550-07590] and #2 [MAP_RS08795-MAP_RS08875] was confirmed in the Telford and the new closed JIII-386 genome. In addition, the adjacent homolog of deletion #2 region MAP_RS08880 is absent in JIII-386 and Telford but present in S397 (MAPS_RS22430). Most genes of the three proposed deleted regions in MAP are present in MAH and MAA genomes; only six genes of deletion #2 region were absent in MAA.

Only JIII-386 is characterized by deletion of genes from LSP^A^11 [MAP_RS20965-20990 in K-10] but these genes are also missing in other MAP-S type strains, for example in porcine MAP sheep type strain LN20 from Canada [38]. Genes of LSP^A^11 were detected in Telford, S397, MAH104, and MAA RCAD0278

Figure 8 summarizes all of the detected presence and absence differences, frameshifts and lack of annotations of these genomic regions along with the proposed MAP-S specific deletions for all 23 compared genomes. This table clearly shows that LSP*^S^* regions I–IV are present, and the region of deletions #1 and #2 are MAP-S specific. In the contrary, deletion s∆−1 is not specific for all MAP- S strains; genes of this region are only missing in strain S397. However, there was no additional definitive MAP-S/type I and type III genome sequence (region) detected—unique for one of the MAP-S subtypes. LSP*^S^* IV (MAV14, [38]) is completely absent in all MAA genomes and a large part of LSP*^S^* I. Strain MAH 104 shows the highest number of differences within the MAH group, confirming its unique position in the phylogenetic tree most closely related to MAA strains. Identified differences concerning loss of individual genes or gene clusters suggest processes involved in evolution. Furthermore, the identified frameshifted genes or gene deletions detected in individual genomes could be the result of local evolution of MAC from different regions around the world.

### 3.13. Core and Pan Genome Analysis

The search for gain and loss of individual genes aims at reveal steps of evolutionary processes including the adaptation of the different MAP subtypes in specific host environments. For this purpose, unique and common protein coding genes (CDS) of genomes belonging to the phylogenetic clusters in Figure 7 were identified. Different features of EDGAR software were applied.

Figure 9 presents the results of different comparative analyses as Venn diagrams, especially focusing on the different MAP subtypes. In these Venn diagrams, the genes of the core-genome (A–D) or pan-genome (E–H) of individual MAP subsets are compared to the gene content of individual MAP genes or other subsets. These diagrams show the numbers of common genes in the intersection areas and the numbers of unique CDS (singletons) in the outer single-colored areas. The core genes are the reciprocal best hits of common orthologous genes between the compared subsets of genomes. These analyses focus on common genes and omit genes which are frameshifted or interrupted (pseudogenes). Moreover, genes shown here as individual genes are genes for which no clear pairwise orthologues could be identified. The true number of individual genes, specific for only one genome or one individual subset (strict singletons), requires a more stringent calculation, and this was conducted by the respective interface of EDGAR software.

Venn diagrams (A) and (E) present results of MAP-S/Type I and Type III strain comparisons based on core (A) and pan (E) genomes of JIII-386 (complete genome), S397 (draft genome) and Telford (complete genome). The differences concerning (i) the common genes and (ii) the singletons of Telford between both diagrams illustrate that a large number of accessory genes in JIII-386 and S397 genomes are homologues to many genes in Telford. The comparison of Telford with the core genome of MAP-C and of MAP-S/type III reveals, as expected, that most common genes of two groups are shared among MAP-S/type I and type III genomes (B). A high number of common genes (n = 240) between MAP-C core and Telford only (B) are homologous to accessory genes of JIII-386 and S397 pan genome, resulting in only 31 common genes between MAP-C pan and Telford only (F). About 60% of 392 genes common among MAP-S/type III core genome and Telford (B) are homologous to accessory genes of MAP-C pan genome resulting in 160 common genes between MAP-S/type III pan genomes and Telford (F).

The comparison of MAP-S core with MAP-C core genome resulted in nearly identical singleton numbers for both groups (C). Venn diagram (C) compared with (G) additionally revealed that many accessory genes of individual strains of one group are homologous to genes within the other group (MAP-S versus MAP-C). The comparison revealed that more accessory genes in MAP-S homologues genes were found in the MAP-C pan genome than contrary (G).

As expected, in all comparisons (A–H), the number of common orthologous genes is higher if the analysis is based on the pan-genome of subsets than on the core-genome. This rise ranged between 8% (A versus E), 16% (B versus F), 19% (C versus G), and 32% (D versus H) depending on the number of genomes and on which genomes and subspecies were included.

Venn diagrams (D) and (H) depict the close relationship of all included *Mycobacterium avium* complex groups. These results, which are based on analyses of core as well as on pan-genome subsets, reveal many overlaps with common genes among the individual comparisons of diverse subspecies and subgroups. However, the three MAP-S strains share a similar number of genes with only MAH and MAA group in the core (D), but nearly twice as many in the pan (H) genome analyses, compared with the MAP-C strains. This is an additional indication for the higher diversity of MAP-S genomes and suggests a slightly stronger specialization of MAP-C relative to MAP-S strains. Apart from that, the pan-genome based analysis (H) revealed about 1525 unique accessory genes in the five included MAH genomes in contrast to only 236 and 225 such singletons in 14 MAP and three MAA genomes. This larger number of unique genes in MAH strains relative to the other subspecies reflects once more the higher diversity of opportunistic MAH organisms adapted to survive in different environments compared with the both obligate pathogenic subspecies MAP and MAA adapted to specific hosts [6,116], which have reduced genome sizes.

Table 7 presents the number of pan genes, core (common) genes and accessory (unique) genes of MAP-S, MAP-C, and all MAPs in comparison to MAH and MAA genomes. Here the pan genome of the three MAP-S/type I+III genomes contained 4482 total genes. This gene number is nearly identical to that of the pan genome of eleven of the twelve MAP-C genomes (n = 4403 genes), supporting the presumed higher diversity of MAP-S genomes. The pan genome of all MAP strains together (n = 14) comprise in total 4726 genes, subdivided in 78% core genes and 22% accessory genes. Despite a different MAP genome panel (including 12 MAP-C genomes and one MAP-S genome) and different analyses software, comparable data were published very recently by Bannantine et al. [35]: a pan genome with 4669 genes consisting of 80.3% of core and 19.7% accessory genes across all MAP strains.

Furthermore, the number of genes in the pan genome of MAH and MAA is also presented in Table 7. A much higher number of genes was detected in the pan genome of MAH (6009 genes) compared with that of the MAA strains and MAP. Based on five MAH genomes, 34% belong to the accessory genome in MAH, but only 9 to 11% of the pan genomes belong to accessory genome in the obligate pathogens MAA, MAP-C, and MAP-S. Non-MAP genomes together were comprised of 29% accessory genes, and this number grew to 60% when twice as many MAH genomes were characterized in the above mentioned study [35].

To further address the question of differences in presence, absence and function of individual genes between the different subgroups, strict singletons were identified using the appropriate interface of EDGAR software. Strict singletons shown here are genes without any relevant hit against any other genome/subset in the analysis. Frameshifted genes (and other pseudogenes) are again not included.

First, strict singletons in MAP-S/type III strains in comparison with Telford and the reverse were calculated. Altogether 37 genes (see Appendix A), present in MAP-S/type III strains JIII-386 and S397 core genome, were absent or frameshifted (*, n = 3) in MAP-S/type I strain Telford. Within the individual genomes, the identified singletons were found to be located separately across the entire genomes, with no indication of clustering. These genes encoded different MFS transporters, five different transferases, a cytochrome bc complex subunit, an RND family transporter*, an Mce family protein*, mbtD*, and nine hypothetical and other proteins. Furthermore, eight of these singletons were also absent in the MAP-C core genome, including genes for three different transferases, a salicylate synthase and a type II secretion system F family protein. Only four were also absent in the MAP-C pan-genome. However, most often genes encoding cytochromes (components of respiratory electron transport chain) or proteins involved in membrane transport could be associated with type specific virulence characteristics. The 27 strict singletons of Telford (MAP-S/type I) that were not present in MAP-S/type III (JIII-386 and S397) pan genome (all listed in Appendix A), include genes encoding two PE family proteins, a transcriptional and a PAS regulator protein, a cell surface protein, a protein translocase subunit, some proteins with other functions, and 17 unknown hypothetical proteins. Twenty of these singletons were also absent from MAP-C core, seven absent from MAP-C pan genome (shown in Appendix A). In contrast, 275 unique genes were detected in Telford if compared with the much smaller core but not with the pan genome of MAP-S/type III strains. These were genes encoding cytochromes (cytochrome 450) or related subunits (n = 5) and type VII secretion proteins (n = 5), which are presumably linked to mycobacterial virulence [117]. However, the predicted functions of the singletons in Telford (type I) as well as in type III strains do not belong to a single metabolic pathway.

Strict singleton genes of the MAP-S core genome (JIII-386, S397, and Telford), not present or frameshifted in MAP-C pan genome, belong to the 85 genes of the LSP^S^s I-IV regions (see Figure 8 and Appendix A) and include 35 additional genes. These 35 genes (see Appendix A) comprise 24 pseudogenes in MAP-C (mostly frameshifted) and 10 gene sequences which are absent in all MAP-C. In each case three genes belonging to two clusters in MAP-S core were also present in MAH 104, suggesting that they were deleted from the ancestor of MAP-C (CDQ89_02225-02235; CDQ89_10290-10300). These two small deletions (each of 2 kb) in MAP-C have not been detected in previous studies. Altogether, the identified singletons encode for five different transcriptional regulators, proteins that are involved in different transport mechanisms, a membrane protein, four hypothetical proteins and others. About 218 more singletons were identified in MAP-S core compared with MAP-C core instead of the MAP-C pan genome, emphasizing the high number of accessory genes in MAP-C pan group. This enlarged MAP-S core gene group included unique genes encoding for a cytochrome P450 and three type VII secretion proteins (EccC and EccE membrane components), which are also part of the MAP-C pan genome.

In the MAP-C core genome (n = 11) compared with MAP-S core genome (n = 3) 252 strict singletons were identified in addition to genes of MAP-S core specific deletions #1 and #2. As above, these singletons (see Appendix A) include genes encoding for nine cytochrome 450 and cytochrome subunit proteins, and also for two type VII secretion proteins (here type EccB and EccD), many other proteins, and 78 hypothetical proteins. Remarkably, only 24 of these MAP-C singletons belong to 275 genes that were listed by Marri et al. [111] as possibly being acquired by lateral gene transfer (LGT) in MAP.

Strict singleton analysis in the MAP-C core genome in comparison to MAP-S pan-genome identified 36 genes (Appendix A), two of which encode for different cytochrome 450 proteins (MAP_RS00055 and MAP_RS06975), which have no detectable homolog in the MAP-S pan genome. In contrast, no singletons were identified in MAP-S core genome encoding cytochromes that were absent also in MAP-C pan genome.

The final analysis included a search for common presumed virulence genes, which are present in MAP & MAA (core), but absent or frameshifted in MAH core and pan genomes. The strict singleton analysis resulted in 156 genes (comparison with MAH core, Appendix A) comprising genes encoding two cytochromes, two type VII secretion proteins, two chemotaxis CheB proteins, different membrane and transport proteins, a Mce family protein, a polysaccharide biosynthesis protein, and others. All these genes could be associated with many virulence characteristics including biofilm formation and chemotaxis. It is interesting, that in this subset there are also 21 genes that belong to the proposed genes originating from LGT [111]. However, there are only three strict singletons in MAP and MAA core genome compared with the large MAH pan genome: MAP_RS06255, MAP_RS12970 and MAP_RS19560 (encoding a dehydratase, hypothetical protein, and IS110 family transposase). This adds 36 other genes, not identified in the previous analysis, which were possibly acquired by LGT [111], and which were found in strict singleton analyses of MAP core genome in comparison to MAA & MAH core and pan genome.

Results of the current core and pan genome analyses with MAP, MAA, and MAH genomes of different worldwide origin emphasize the close relationship among strains of *M. avium* subspecies. In agreement with results described elsewhere [35], results from this work show that the larger MAH genomes harbor a higher number of accessory genes within their pan genome, undoubtedly important for their ubiquitous occurrence and more facultative pathogenic nature in comparison to the host adapted obligate pathogen organisms of MAP or MAA subspecies [5]. Furthermore, this higher diversity of MAH strains was also related to a higher number of hsp65 sequevars and SNPs, a higher number of proposed recombination sites, and occurrence of plasmids [35,74,118]. In the current study, the detection of various strict singletons belonging to a group of genes previously identified as having been acquired by horizontal gene transfer [111] within the core genome of the different MAP strain subsets suggests the repeated exchange between the progenitor(s) of present strains with other bacteria in their environment during evolution or adaptation to their specific environmental niches or hosts. To understand the specific virulence characteristics or other phenotypic features of *M. avium* subspecies, MAP types and subtypes, and their evolution, it is important to consider not only the presence or absence but also the functionality of genes (intact, frameshifted, or interrupted) which are likely to cause or regulate these features.

## 4. Summary

A major goal of this study was to close the draft genome of the German ovine MAP strain JIII-386 using the relatively novel Nanopore Technology and polish this closed genome with deep whole-genome shotgun Illumina paired-end sequencing data. Based on this new closed MAP-S/type III strain JIII-386 and the recently published complete genome sequence of MAP-S/type I strain Telford from Australia, the present study aimed to close knowledge gaps concerning genomic differences between MAP-S subgroup I and III, and MAP-S and MAP-C (type II) strains, and to reveal new insight into MAP genomics and evolution. Although all bioinformatic methods used for the genome-wide comparisons are optimally applied to closed complete genome sequences, the high-quality draft genome of type III strain S397 from U.S. was also analyzed to increase the number of MAP-S genomes. In addition to MAP-S genomes, the study included twelve MAP-C, three MAA and five MAH genomes, all closed and annotated by NCBI PGAP.

In the first part of the study, general genomic features of the newly closed JIII-386 were described and compared with the draft genome of JIII-386 and S397 and the completed genomes of Telford and K-10. MAP-S/type I strain Telford is the largest genome here comprising more CDS and genes with predicted function than MAP-S/type III and MAP-C genomes. The Shine–Dalgarno sequence motif 5′-AGCTGG-3′ suggested to be characteristic for Mycobacteria [34], was confirmed for new closed MAP-S strains, and also for different other MAP-C and for MAA strains which may be helpful in future bioinformatic studies. The structural comparative alignments (MAUVE plots) among JIII-386, Telford and K-10 genomes and also the distribution of corresponding genomic islands clearly reveal the mosaic nature of MAP genomes with different type and subtype. Specific genetic elements such as CRISPER loci, fragments of putative prophages and like regions, IS elements, and genomic islands were identified, and in many cases confirmed by using multiple approaches to avoid misrepresentations. The comparison of JIII-386, Telford, and K-10 revealed many conformities. These different MAP-subtype genomes contained no functional CRISPR-Cas region and only one large potential fragment (17.7 kb) of a prophage genome. JIII-386, Telford and K-10 share many secondary metabolite compound clusters. None of the different MAP type genomes harbored any typical antibiotic resistance genes, but all of these strains harbored a homologous gene for a known ABC-type multidrug transport system that is involved in drug resistance in *M. tuberculosis*. Comparisons of selected putative virulence genes and gene clusters (*mce*, *mmpL*, *rnd*, and PE/PPE genes; and Mycobactin gene cluster) show a high conformity but also differences concerning the presence, the length, the sequence and the completeness of individual genes as summarized below. Comprehensive core and pan genome analysis based on the enlarged genome panel, revealed the large number of highly conserved common genes but also unique genes (accessory genes) probably acquired by horizontal gene transfer in different MAP-types.

The phylogenetic tree based on core genome of 24 *M. avium* genomes clearly illustrates the distinct grouping of MAP-S and MAP-C as well as MAP-S type I and III sub-clusters. In addition, the SNP analyses conducted for this work support the tree results. The study confirmed the results of earlier previous reports that major genetic differences between MAP-S and MAP-C group include genes of the known large sequence polymorphisms (LSP*^S^*s) regions I–IV which are absent in MAP-C, and genes of deletions #1 and #2 which are absent in MAP-S, which could potentially account for the major type specific phenotypic features. Additional differences were uncovered. Type I and type III MAP-S genomes showed a higher structural diversity in contrast to MAP-C genomes, which exhibited a strong synteny of chromosomal structure. The number of specific IS elements (IS*900*, IS*1311*, IS3 family, and IS*Map02*) differed; in total more IS elements were detected in MAP-S than in MAP-C genomes. Many more hints of differences between MAP-S and MAP-C genomes were confirmed and new ones identified due to detection of known virulence-associated genes. Both MAP-S subtype strains harbored two mce CDS that were outside of detectable mce operons, whereas MAP-C genomes harbored only one. The current analysis confirmed that the MAP-C strains includes an additional mmpL gene that was absent in both MAP-S genomes. There were individual PPE genes which were lost in one or the other MAP-type. MAP-type specific differences in the Mycobactin gene cluster were determined that could possibly explain sub-type specific differences in growth characteristics. Compared to K-10 (MAP-C), in type I and III strains of MAP-S the gene *mbtA* is not truncated, the gap between *mbtA* and *mbtJ* is larger (53 kb vs. 19 kb), and *mbtC* is interrupted. Furthermore, the gene *mbtE* gene is shorter and as with *mbtA* shows only up to 40% sequence identity to the same genes in K-10. Moreover, MAP-C genomes include an additional secondary metabolite gene cluster. The core genome analyses uncovered many unique genes in MAP-S or MAP-C core genome, absent or frameshifted in the core or the pangenome of the other group. These include genes encoding cytochrome 450, different cytochrome subunits and different type VII secretion proteins, which are likely to be associated with virulence. Remarkably, in the core genome of MAP-C, two genes were identified encoding for cytochrome 450 proteins (MAP_RS00055 and MAP_RS06975), without any homolog in the MAP-S pan genome. MAP-S genomes contained two additional small gene clusters of 2 kb each with three genes, that were absent in MAP-C genomes possibly lost during MAP-C evolution.

Another main objective was to clarify which genetic differences assist with subdivision of MAP-S into type I and type III subgroups. In summary, the following results support the subdivision: (1). Structural comparative alignments (MAUVE plots) show a relatively strong synteny among both MAP-S/type III genomes (JIII-386 and S397) but much more rearrangements and inversions between type III and Telford (type I) genomes. (2). The SNP analysis revealed a lower number of SNPs in comparison of the two type III genomes than in a comparison of type III and type I genomes. (3). The genome of type I strain Telford is larger; contains more IS elements and a higher number of GIs were identified in Telford than in type III strain JIII-386 suggesting different evolutionary adaptation processes. (4). Within the mycobactin cluster, JIII-386 and S397 showed different sizes for the gap between *mbtJ* and *mbtI* when compared with Telford (5). The search for unique genes in type III strains (JIII-386 and S397) and or Telford, revealed 37 and 27 genes respectively, and these were predicted to encode several virulence associated proteins. All results of these comparisons will need to be re-examined when more closed MAP-S type genomes become available.

Because the divergence of MAP subtypes from their common ancestor is a relatively recent event in the evolution of *M. avium*, there are fewer genomic differences relative to other MAC bacteria. Results from the current and previous studies hint at the evolutionary origin of MAP-S/type I, type III, and MAP-C/type II subgroups in relation to MAH and MAA. Results of the current analyses confirmed that MAP-S strains are more diverse than MAP-C strains. In addition, the larger genome size of Telford, and the higher number of IS elements and GIs, compared to JIII-386, suggest a possible earlier emergence of type I than type III. The phylogenetic tree presented here suggested a slightly earlier emergence of MAP-S strains from a proposed common progenitor than MAP-C strains. MAP-C strains seem to have emerged more recently, possibly because of intensive cattle farming combined with an increasing MAP contamination of the environment as well as distribution across the world by intensive livestock trade. MAA and MAP are pathogens that are well adapted to their hosts, but MAA is more closely related to MAH than to MAP. The understanding of the emergence of obligate human and animal pathogens possibly from more ubiquitous occurring, facultative pathogenic members of the MAH group is very important in a world which is burdened by overpopulation and profound changes to the environment including those occurring in pathogenic and non-pathogenic bacteria.

In conclusion, here we compared genome structure, virulence associated genes, conserved genomic island and core genomes of available complete or nearly complete MAP genomes and close MAC relatives. We found that the few available MAP type S genomes are distinct from MAP C types. The genomic variations among MAP-S type III strains collected from different geographic locations were small enough to form a close cluster and the genomic variations among the strains in MAP-S types I and type III are big enough to form two distinct groups. Major limitations of these analyses are the paucity of differences due to the very slow tendency of MAP to acquire genomic changes relative to other bacteria and a lack of representative S type references. These groupings will become more robust as more type I and type III S genomes from different parts of the world become available for use as references.

However, the finished S type genome sequences of JIII-386 (type III) and Telford (type I) that were scrutinized here provide a valuable resource for further *M. avium* research. The comparison of MAC genomes from the current study provide a reference of MAP specific similarities and differences that may help to improve our understanding of mycobacterial pathogenicity and evolution.

## Figures and Tables

**Figure 1 microorganisms-09-00070-f001:**
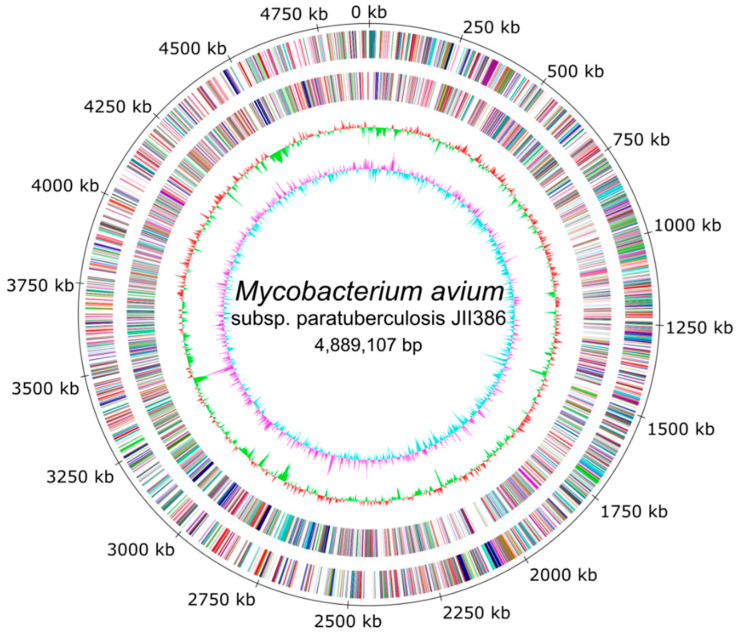
Circular map of sheep isolate JIII-386. The circles represent (from innermost to outermost) GC skew and GC content (in blue and green: values below the average; in purple and red: values above the average), predicted protein coding sequences (CDS) transcribed anticlockwise (inner part) or clockwise (outer part), and genomic position in kb. The CDSs are colored according to the assigned clusters of orthologous groups (COG), [67]. The gene *dna*A is localized at the 0 kb locus.

**Figure 2 microorganisms-09-00070-f002:**
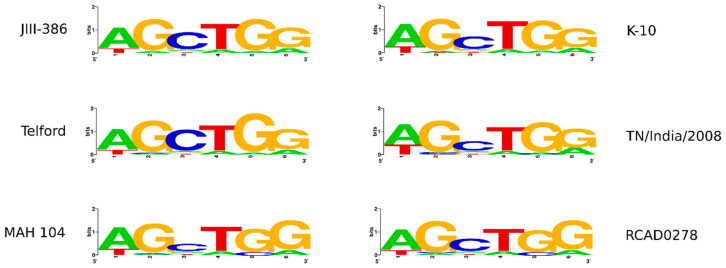
Shine–Dalgarno sequence motifs of six investigated *M. avium* strains.

**Figure 3 microorganisms-09-00070-f003:**
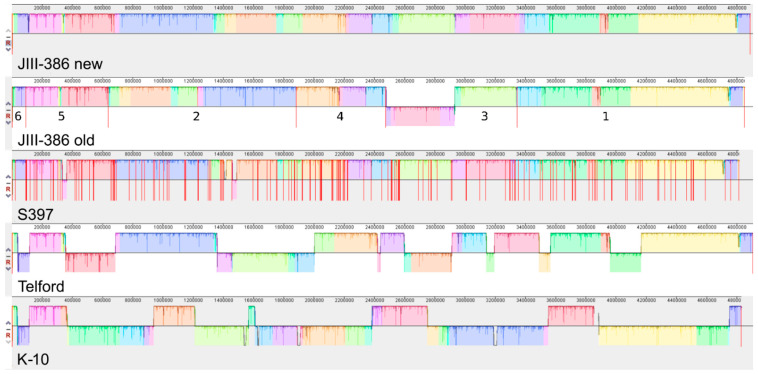
Comparison of new and old versions of JIII-386, S397, Telford, and K-10 using MAUVE 2.4 [53]. The scale is in base pairs. All genomes are compared to a consensus genome generated by Mauve. Identical colored blocks indicate homologous regions, which are internally free from genomic rearrangements. Blocks below the centerline are aligned reverse complementary. Darker lines within each box represent regions that differ in comparison to the consensus sequence, with longer lines being more different than shorter lines. Long red lines demonstrate the gaps in JIII-386 old and S397 draft genomes. The six scaffolds in JII-396 are numbered in descending size order. The start of the *dnaA* gene is localized at 0 kb.

**Figure 4 microorganisms-09-00070-f004:**
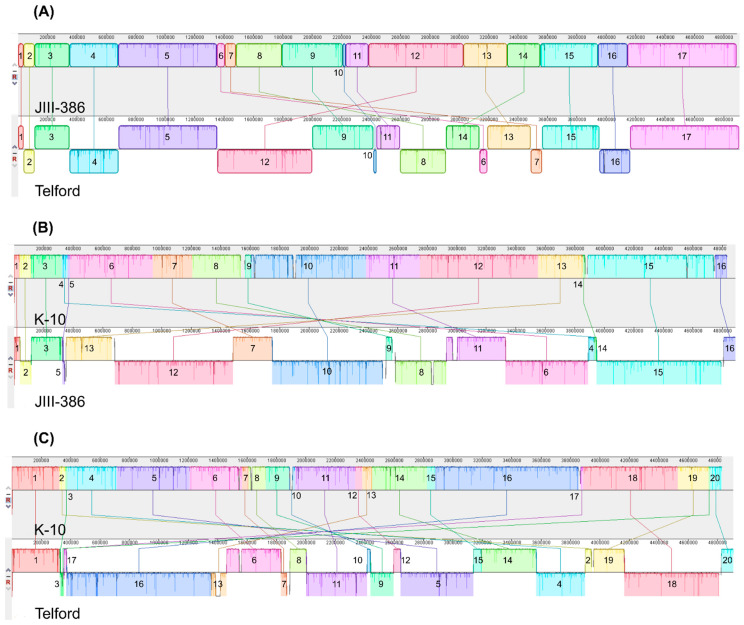
Pairwise comparison of MAP JIII-386, Telford and K-10 genome sequences using Mauve 2.4 [53]. The scales are in base pairs. The *dna*A gene is positioned at 0 kb. Homologous segments with similar DNA profile among the strains are represented by identically colored boxes. A number was assigned to each segment. Lines connect identically colored boxes. Segments below the centerline indicate inverse orientation. (**A**) MAP JIII-386 and Telford (MAP-S/type III and I): Eight of the 17 total segments are revers complementary orientated including rearrangements of four segments (~35% of the total genome size). In addition, two segments with the same orientation were rearranged. (**B**) MAP K-10 and JIII-386: Seven out of 16 total segments are revers complementary orientated including three rearrangements (~68% of the total genome size). Three segments that were not in the reverse order were found at different relative chromosomal locations. (**C**) MAP K-10 and Telford: Nine of 20 total segments are reverse complementary orientated including three rearrangements (~65% of total genome size). In addition, six segments, which were not in the reverse complimentary order, were found in different chromosomal positions.

**Figure 5 microorganisms-09-00070-f005:**
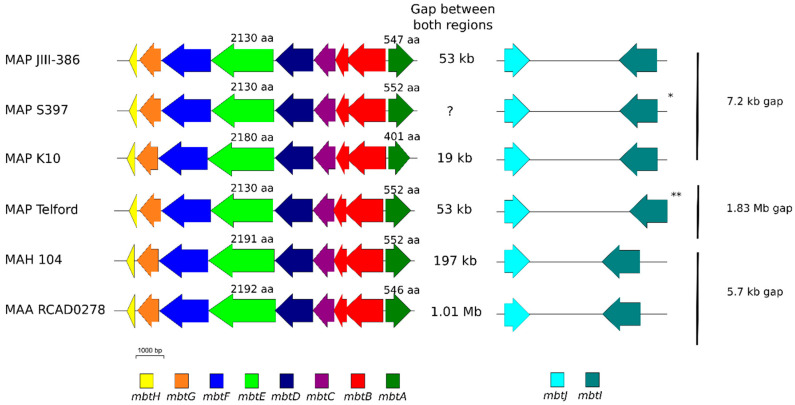
Mycobactin cluster in MAP-S/type III strains JIII-386 and S397, MAP-C strain K-10, MAP-S/type I strain Telford, MAH 104 and MAA RCAD0278. Identical colors matched to identical genes. Gene lengths are shown for *mbtA* and *mbtE* because of existing differences. MAP-S strains share a gap of identical size between *mbtA* and *mbtJ* genes in contrast to the other strains. * Predicted size of gap between *mtbJ* and *mbtI:* genes that are located on different contigs; ** Because of a rearrangement in Telford, *mbtJ* and *mbtI* genes are far apart from each other.

**Figure 6 microorganisms-09-00070-f006:**
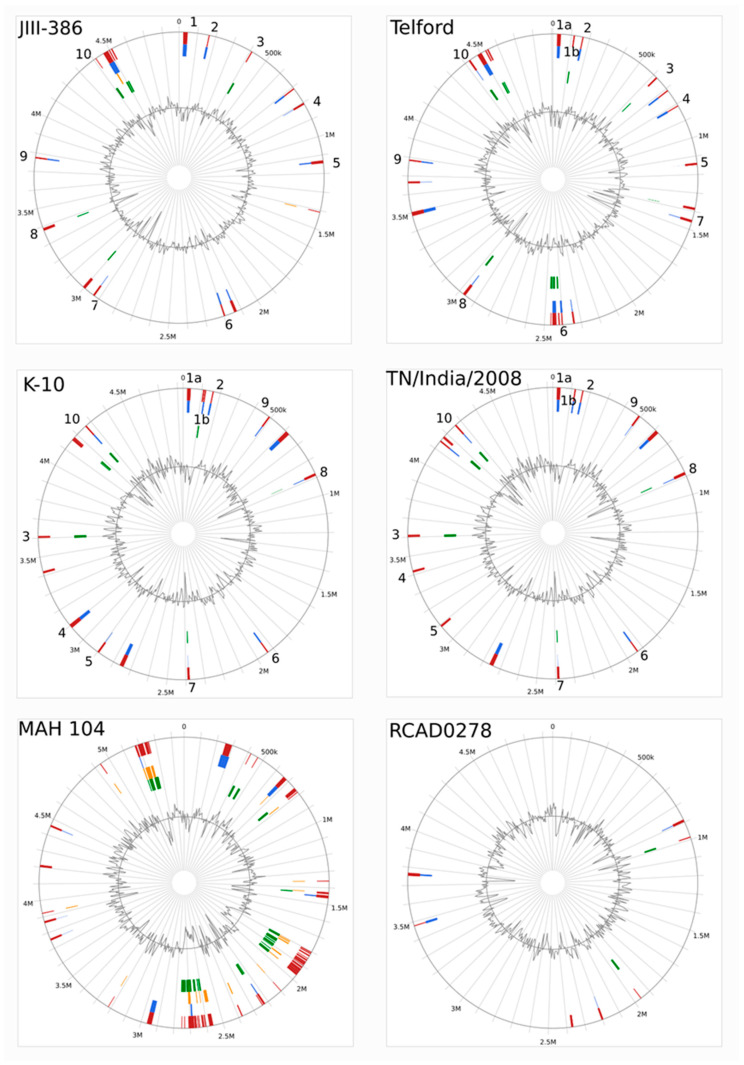
Plot of genomic islands in circular maps of 6 *Mycobacterium avium* strains using Island Viewer 4. This method uses both Island Pick (green bares), Island Path DIMOB (blue bares), an integrative approach (red bares), as well as manual inspection to identify differences in external annotation. The inner most circle in the figure represents the GC content, with values below the average on the inside and values above the average on the outside of the circle. Genomic positions are represented in bp. Represented strains are: JIII-386 (MAP-S/type III), Telford (MAP-S/type I), K-10 and TN/India 2008 (MAP-C), MAH 104 (*M. avium* subsp. *hominissuis*), and RCAD0278 (*M. avium* subsp. *avium*). Labels numbered 1 to 10 correspond to genomic islands detected in MAP-S and MAP-C strains (see Appendix A) and are based on GIs in JIII-386: 1 = J1a, 2 = J2, 3 = J3, 4 = J5, 5 = J6, 6 = J8, 7 = J9, 8 = J11, 9 = J12, 10 = J13).

**Figure 7 microorganisms-09-00070-f007:**
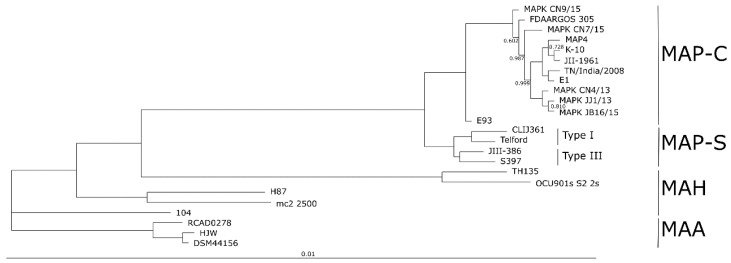
Phylogenetic tree of 24 selected *Mycobacterium avium* (*M. a*.) complex strains. These strains belong to the different subspecies *paratuberculosis* (MAP), *hominissuis* (MAH), and *avium* (MAA), originating from different locations and hosts (see Table 1). The phylogenetic tree was constructed from concatenated core gene alignments (2807 genes per genome, 67,368 genes in total) based on the approximately Maximum-Likelihood method FastTree 2.1 [64] using the Jones-Taylor-Table [113]. To account for the varying rates of evolution across sites, the CAT approximation with 20 rate categories was used. The core has 923,637 AA-residues/bp per genome, 22,167,288 in total. To estimate the branch support, SH values were calculated by FastTree 2.1 [64]. The majority of branches showed a perfect support value of 1000, branches with lower SH values are indicated in the tree.

**Figure 8 microorganisms-09-00070-f008:**
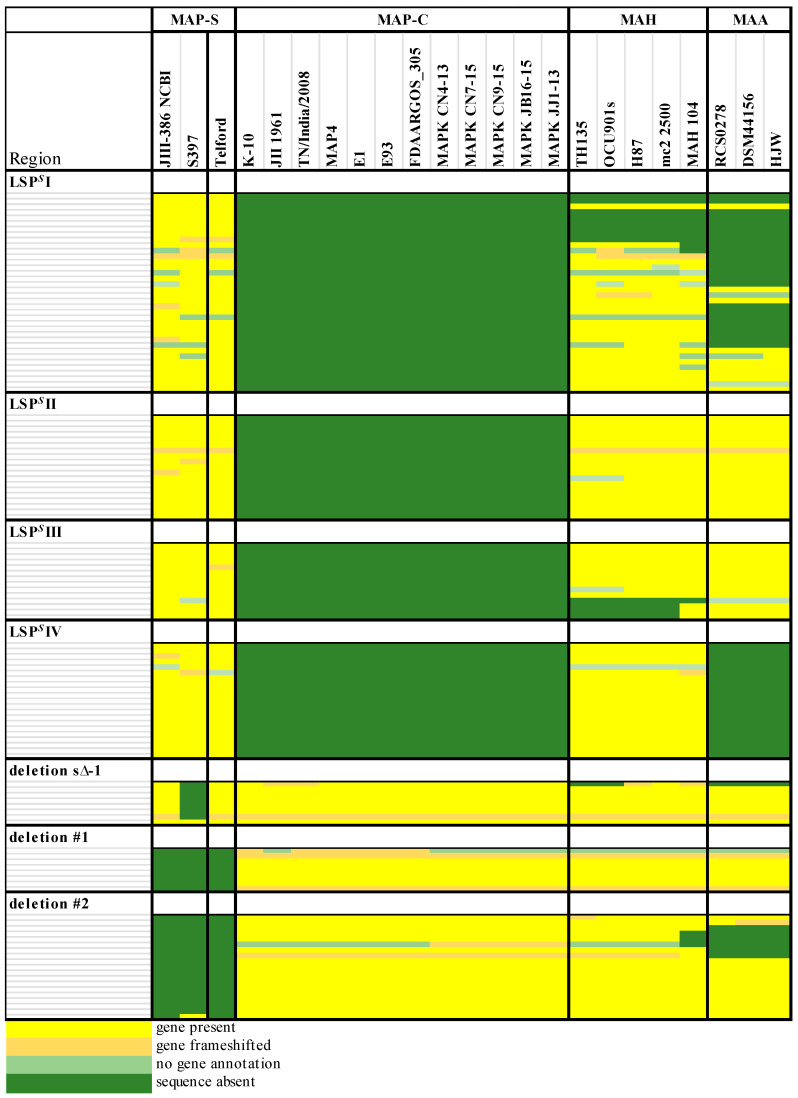
Presence and absence of genes belonging to MAP-S specific LSP*^S^*s I–IV and predicted deletions within 23 different *Mycobacterium avium* complex (MAC) genomes belonging to MAP-S, MAP-C, MAH and MAA. All protein coding genes (CDS) belonging to these regions and their predicted functions are presented in Appendix A.

**Figure 9 microorganisms-09-00070-f009:**
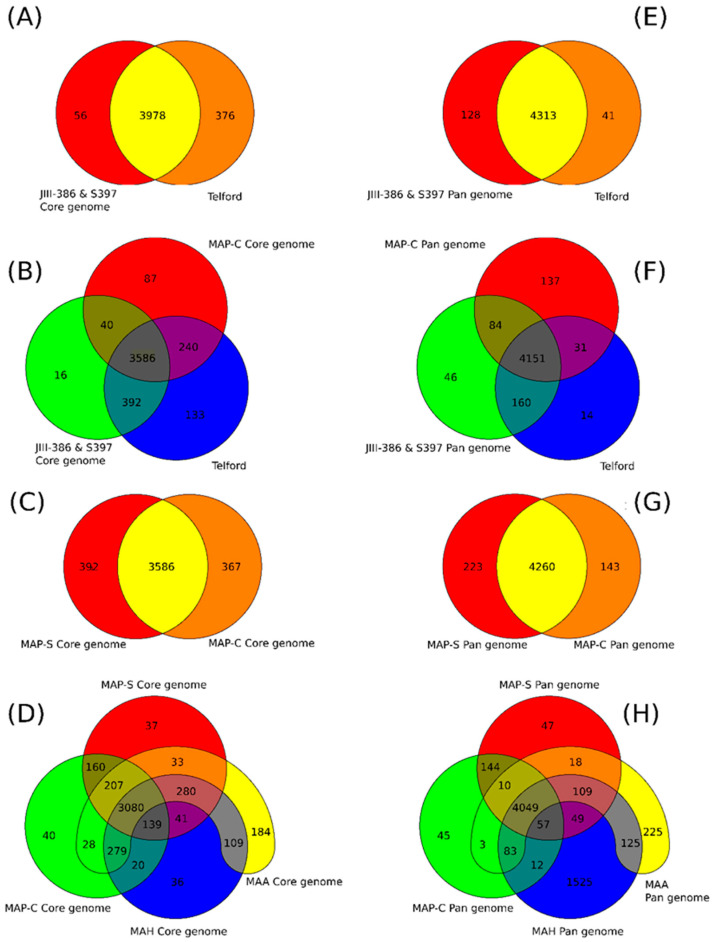
Venn diagrams show results of protein coding gene comparisons of individual genomes and sets of core (**A**–**D**) and pan (**E**–**H**) genomes. Diagrams illustrate the number of common genes shared by the specific subsets of genomes shown within the intersections (yellow or mixed) and the number of singleton genes (red, orange, green, blue, and yellow areas). The MAP-S group includes JIII-386, S397, and Telford, and the MAP-C group is comprised of 11 genomes (Table 1, without E93). Genomes belonging to MAH group (n = 5) and MAA group (n = 3) are listed in Table 1. The core and the pan genomes of individual subgroups were calculated with EDGAR software. The pan genome of MAH group was calculated without plasmids.

**Table 1 microorganisms-09-00070-t001:** *Mycobacterium avium* (M. a.) strains (n = 24) of different geographic and host origin whose NCBI reference genome sequences and annotations (all annotated by the NCBI PGAP pipeline) were involved in this study.

Organism	Type	Strain	Accession No.	Level	Origin	Host
MAP ^(1)^	S(III)	JIII-386	NZ_CP042454	closed	Germany	sheep
	S(III)	S397	AFIF00000000.1	draft	U.S.	sheep
	S(I)	Telford	NZ_CP033688.1	closed	Australia	sheep
	S(I)	CIJ361 ^(4)^	AFNS00000000.1	draft	Australia	sheep
	C	K-10	NC_002944.2	closed	U.S.	cattle
	C	JII-1961	NZ_CP022105.1	closed	Germany	cattle
	C	TN/India/2008	NZ_CP015495.1	closed	India	cattle
	C	E1	NZ_CP010113.1	closed	Egypt	cattle
	C	E93 ^(4)^	NZ_CP010114.1	closed	Egypt	cattle
	C	MAP4	NZ_021200.1	closed	U.S.	human
	C	FDAARGOS_305	NZ_CP022095.2	closed	U.S.	cattle
	C	MAPK_CN4/13	NZ_CP033910.1	closed	S.K. ^(5)^	cattle
	C	MAPK_CN7/15	NZ_CP033428.1	closed	S.K. ^(5)^	cattle
	C	MAPK_CN9/15	NZ_CP033427.1	closed	S.K. ^(5)^	cattle
	C	MAPK_JB16/15	NZ_CP033911.1	closed	S.K. ^(5)^	cattle
	C	MAPK_JJ1/13	NZ_CP033909.1	closed	S.K. ^(5)^	cattle
MAH ^(2)^		104	NC_008595.1	closed	U.S.	human
		H87	NZ_CP018363.1	closed	U.S.	water
		mc2 2500	NZ_CP036220.1	closed	U.S.	human
		OCU901s_S2_2s	NZ_CP018014.2	closed	Japan	human
		TH135	NZ_AP012555.1	closed	Japan	human
MAA ^(3)^		DSM 44156	NZ_CP046507.1	closed	unknown	Gallus gallus
		RCAD0278	NZ_CP016396.1	closed	China	duck
		HJW	NZ_CP028731.1	closed	China	cattle

^(1)^*M. a.* subsp. *paratuberculosis*; ^(2)^
*M. a.* subsp. *hominissuis*; ^(3)^
*M. a.* subsp. *avium*; ^(4)^ genome sequences used for calculation of the phylogenetic tree only; ^(5)^ South Korea.

**Table 2 microorganisms-09-00070-t002:** General genome features of different *M. avium* subsp. *paratuberculosis* (MAP) strains.

Feature	New JIII-386	Old JIII-386	S397	Telford	K-10
type	MAP-S(III)	MAP-S(III)	MAP-S(III)	MAP-S(I)	MAP-C
Accession Number	CP042454	NKRI010000000 ^1^	AFIF000000000 ^1^	CP033688.1 ^1^	NC_002944.2 ^1^
Assembly Status	Closed	6 scaffolds	176 contigs	Closed	Closed
Size (bp)	4,889,107	4,850,274	4,813,711	4,907,428	4,829,781
GC content (%)	69.24	69.16	69.31	69.24	69.30
Total number of genes	4631	4563	4644	4700	4562
rRNA operons	1	1	1	1	1
tRNAs	47	46	46	46	46
Protein coding genes (CDS)	4578	4511	4593	4648	4510
Genes with predicted function	3183	3178	3194	3342	3161

^1^ Genome was exported from NCBI RefSeq database.

**Table 3 microorganisms-09-00070-t003:** Number of insertions and their total length in JIII-386, Telford, and K-10.

	JIII-386	Telford	K-10
JIII-386	-	181 (7497 bp)	327 (160,988 bp)
Telford	493 (25,817 bp)	-	319 (141,552 bp)
K-10	677 (101,662 bp)	397 (63,906 bp)	-

The first column includes individual strains (in bold), which has insertions in comparison to JIII-386 (Column 2), Telford (Column 3) and K-10 (Column 4).

**Table 4 microorganisms-09-00070-t004:** Number of specific elements in different MAP genomes.

Name	JIII-386	S397	Telford	K-10
CRISPER loci (questionable/evidence level 1)	0 (7)	0 (8)	0 (3)	0 (8)
Fragments of putative prophages or like regions ^(a)^	1	0	1	(1)
IS elements or transposase family CDS	55	n. d. ^(b)^	60	49
Mce family protein CDS (gene) cluster	7 + 1 ^(c)^	7	8	8
MmpL and rnd protein CDS (gene)	16	16	16	17
PPE family protein CDS (gene)	35	35	35	34
PE family protein CDS (gene)	8	8	8	8 + 1 ^(c)^
PE_PGRS family protein CDS (gene)	1	0/n. d. ^(b)^	1	1 ^(d)^

^(a)^ Detected by PHASTER prophage finder software; ^(b)^ not exactly detectable (draft genome sequence); ^(c)^ Incomplete; ^(d)^ hypothetical gene, annotation found by sequence comparison using PE_PGRS family protein CDS sequence of JIII-386.

**Table 5 microorganisms-09-00070-t005:** Secondary metabolite clusters in JIII-386 with ≥60% gene similarity to antiSMASH database.

Cluster No.	Position (bp)	Name of Enzymes	CDS	Predicted Product
1	1,234,319–1,271,464	Type 3 PKS ^1^	21	Alkylresorcionol
2	2,929,668–2,995,077	NRPS ^2^	16	Mycobactin

^1^ Polyketide synthases; ^2^ Non-ribosomal polypeptide synthases.

**Table 6 microorganisms-09-00070-t006:** Confirmed MAP-S specific large sequence polymorphisms (LSP*^S^*) regions I–IV. All protein coding genes (CDS) belonging to these regions in individual MAP-S, MAH, and MAA strains are presented in Appendix A.

LSP^S^	Old LSP^S^	Island Size ^∆^	# CDS ^∆^	Present in			
[34]	Included	MAP-C Negative	MAP-S	MAP-S	MAP-C	MAH 104	MAARCD0278
LSP^S^ I	10 CDS of LSP^S^1 */22 of LSP^A^ 4-II **	33,683 bp	32	Yes	Not	Partly	Partly
LSP^S^ Ia	2 CDS of LSP^S^1 *		10	Yes	Not	Not	Not
LSP^S^ Ib	8 CDS of LSP^S^1 */22 CDS of LSP^A^ 4-II **		22	Yes	Not	Yes	Partly (n = 9)
LSP^S^ II	LSP^S^2 + 4 */LSP^A^ 18 **	16,192 bp	19	Yes	Not	Yes	Yes
LSP^S^ III	LSP^S^5 + 7 */GPL **	16,123 bp	14	Yes	Not	Partly	Yes
LSP^S^ IIIa	LSP^S^5 + 7 */GPL **		11	Yes	Not	Yes	Yes
LSP^S^ IIIb			3	Yes	Not	Not	Yes
LSP^S^ IV	MAV-14 **	21,100 bp	20	Yes	Not	Yes	Not

* [33]; ** [38]; ^∆^ Island size and number of included CDS in JIII-386.

**Table 7 microorganisms-09-00070-t007:** Number of genes in the pan genomes of MAP-S, MAP-C, MAH, and MAA. The pan genome is subdivided into core and accessory genomes. The calculation based on analyses by EDGAR, which was also used for results shown as Venn diagrams in Figure 9.

	MAP-S	MAP-C	MAP	MAH	MAA	Non-MAP
	(n = 3)	(n = 11)	(n = 14)	(n = 5)	(n = 3)	(n = 8)
pan gnome	4482	4403	4626	6009	4622	6265
core genome	3978 (89%)	3953 (90%)	3586 (78%)	3984 (66%)	4200 (91%)	4436 (71%)
accessory genome	504 (11%)	450 (10%)	1040 (22%)	2025 (34%)	422 (9%)	1829 (29%)

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
