# Peer review of "Complete Genome Sequence of Ovine Mycobacterium avium subsp. paratuberculosis Strain JIII-386 (MAP-S/type III) and Its Comparison to MAP-S/type I, MAP-C, and M. avium Complex Genomes"

_microorganisms, 2020, doi:10.3390/microorganisms9010070_

Round 1

Reviewer 1 Report

In this paper, the authors analyzed the whole genome of a MAP-S type III strain JIII-386 and compared its genome with the representative strains of other groups. The authors sought to delineate the relationships of MAP-S and MAP-C as well as of MAP-S type I and type III genomes. JIII-386 genome reported in this study was partially sequenced and previously reported. Based on genome analysis, JIII-386 was classified as MAP-S type III. The authors used publicly available MAP genomes of other groups to determine that MAP-S type I and III, can be separated based on the chromosomal organization, total number of SNPs, and INDELs; MAP-S type I and III strains have closer relationship than strains in MAP-C group.

Overall, this study is well done and importantly increases the number of MAP sequences collected in the GenBank. The authors successfully sequenced the entire genome of JIII-386 using a strategy in that large DNA fragments of genome were sequenced with Oxford Nanopore followed by sequence polishing with sequences from short-read sequencing. The sequencing strategy can be applied to study other organisms that lack sufficient genome information. The findings the authors present in this paper provide limited evidence for classifying MAP-S and MAP-C as distinct groups and MAP-S I and III as distinct subtypes. However, I think that the limitations of this study should be more thoroughly explored to further bolster the authors' argument.

Major points

The authors apply results from genome analysis of one strain to make conclusions about the genetic relationships of different MAP groups. The conclusions are established based on the assumptions: (1) the genomic variations among MAP-S type III strains collected from different geographic locations are small enough to form a close cluster; (2) the genomic variations among the strains in MAP-S types I and type III are big enough to form two distinct types. As indicated by the authors in the manuscript, only few MAP-S strains are completely sequenced. The evidence to support these assumptions is lacking. Sequencing more than one strain is necessary to make a generalized conclusion.

The sheep strain JIII-386 was isolated in 2003. How stable is the genome? Can genomic variations be introduced during subculture? Comparing the genomes of the new JIII-386 isolate with the old one showd that the first region in Scaffold 3 of old isolate is in reverse complementary orientation to the corresponding region in the new JIII-386 isolate (Figure 3). Does it suggest that genomic changes can be introduced during passaging of the isolate?

As the authors pointed out in the result section, results of genome annotation vary with different annotation programs. When comparing genome of JIII-386 with other isolates, was the same annotation program used for all compared isolates?

Results are presented along with the discussion in the results section, making the results generated by the study less clear.

Minor points

Line 164 on page 4, “from” is redundant.

As many programs were used for analysis, a table listing pipelines used will make it easier for readers to follow.

Reviewer 2 Report

To author

In order to close the genome draft of MAP strain JIII-386, the authors polished whole-genome shotgun Illumina pair-end sequencing data using nanopore technology. It was analyzed for the genomic differences between the different groups including MAP and MAC based on these closed-type MAP-S/type III genomic sequences.

In my opinion, there are two main points of this manuscript. The first is a complete and carefully closed report of the MAP-S/Type III genome. And secondly, nanopore-technology was used to complete the genomic analysis by completely closing the MAP genome in the present study. Nanopore-technology is a very novel technique for analyzing microbial genomes including MAP. Additionally, the authors have indicated “Nanopore-technology” as the first keyword in this manuscript. Although it is considered one of the main points of this study, it seems that there is a lot of lack of explanation and discussion related to Nanopore-technology in the introduction or discussion of the text. Therefore, I suggest that the authors add an additional explanation for this technique.

This manuscript is very well written, and a well-established methodology for MAP genome analysis is used. Therefore, I suggest it is accepted after a minor revision.

Author Response

Dear Reviewer 2,

Here is our point to point response to your comments and suggestions. We thank for your very helpful and valuable suggestions to improve the manuscript.

Reviewer 2’s Comments and Suggestions for Authors

In order to close the genome draft of MAP strain JIII-386, the authors polished whole-genome shotgun Illumina pair-end sequencing data using nanopore technology. It was analyzed for the genomic differences between the different groups including MAP and MAC based on these closed-type MAP-S/type III genomic sequences.

In my opinion, there are two main points of this manuscript. The first is a complete and carefully closed report of the MAP-S/Type III genome. And secondly, nanopore-technology was used to complete the genomic analysis by completely closing the MAP genome in the present study. Nanopore-technology is a very novel technique for analyzing microbial genomes including MAP. Additionally, the authors have indicated “Nanopore-technology” as the first keyword in this manuscript. Although it is considered one of the main points of this study, it seems that there is a lot of lack of explanation and discussion related to Nanopore-technology in the introduction or discussion of the text. Therefore, I suggest that the authors add an additional explanation for this technique.

We would like to give our sincere thanks to the reviewer for this comment. As recommended, we have added an explanation for the Nanopore-technology in the introduction part (page 3, line 136 to 147). This includes also two additional References as follows:

"Nanopore sequencing technology, developed by Oxford Nanopore, is one of the latest DNA sequencing methods and represents a third-generation approach. Using Nanopore sequencing, a single molecule of DNA is transported through a nanometer large pore (Varongchayakul et al., 2018). These nanopores consist of recombinant proteins embedded in a polymer membrane. A bias voltage is applied across the membrane. Nucleotides that passes through the nanopore create voltage changes that are specific for each of the four nucleotides, enabling the DNA sequence to be read out. Single DNA molecules longer than a megabase can be sequenced using Nanopore, but the resulting sequence has a rather high error rate (usually in the 5-20% range). By means of sequence assembly and polishing of the consensus sequence with high depth Illumina reads, a similar error rate in comparison to short read data can be reached (Kono et al., 2019) but provides sequence over repeat regions and other regions that are not well defined with short read methods."

Quality control statistics for these assemblies are listed in Chapter 2.3.  (pages 4 and 5, lane 193-196).

"Completeness and assembly quality were estimated with BUSCO (v3.0.2, Simão et al. 2015), using the Bacteria-specific single-copy marker genes database (odb9). The BUSCO analysis identified 143 of 148 core bacterial genes. This result emphasizes the completeness and high quality of the genome assembly."

For that the title of Chapter 2.3. was expanded into (see lane 193):

2.3. Base calling, reads processing, assembly, quality control and deposition of new genome sequence.

We hope that these changes will answer your comments.

Sincerely,

On behalf of all co-authors,

Petra Moebius

Round 2

Reviewer 1 Report

The reviewer's concern has been adequately addressed.